# CryoET reveals organelle phenotypes in huntington disease patient iPSC-derived and mouse primary neurons

Gong-Her Wu[1,14], Charlene Smith-Geater [2,14], Jesús G. Galaz-Montoya[1], Yingli Gu [3], Sanket R. Gupte[4], Ranen Aviner[5], Patrick G. Mitchell [6], Joy Hsu [4], Ricardo Miramontes[7], Keona Q. Wang [8], Nicolette R. Geller[8], Cathy Hou[1], Cristina Danita[1], Lydia-Marie Joubert [6], Michael F. Schmid [6], Serena Yeung [4,9], Judith Frydman [5,10], William Mobley[3], Chengbiao Wu[3], Leslie M. Thompson [2,7,8,11,12,15] ✉ & Wah Chiu [1,6,13,15] ✉

Huntington's disease (HD) is caused by an expanded CAG repeat in the huntingtin gene, yielding a Huntingtin protein with an expanded polyglutamine tract. While experiments with patient-derived induced pluripotent stem cells (iPSCs) can help understand disease, defining pathological biomarkers remains challenging. Here, we used cryogenic electron tomography to visualize neurites in HD patient iPSC-derived neurons with varying CAG repeats, and primary cortical neurons from BACHD, deltaN17-BACHD, and wild-type mice. In HD models, we discovered sheet aggregates in double membrane-bound organelles, and mitochondria with distorted cristae and enlarged granules, likely mitochondrial RNA granules. We used artificial intelligence to quantify mitochondrial granules, and proteomics experiments reveal differential protein content in isolated HD mitochondria. Knockdown of Protein Inhibitor of Activated STAT1 ameliorated aberrant phenotypes in iPSC- and BACHD neurons. We show that integrated ultrastructural and proteomic approaches may uncover early HD phenotypes to accelerate diagnostics and the development of targeted therapeutics for HD.

Huntington's disease (HD) is a progressive, fatal neurodegenerative disorder caused by a genetic mutation in the huntingtin gene (HTT)[1]. The mutation is an expansion of a CAG repeat to ~40 and above within the first exon. This yields a mutated Huntingtin protein (mHTT) with an expanded polyglutamine (polyQ) tract that is pathogenic. Disease typically strikes in mid-life, lasting ~10–15 years with an ongoing progression of symptoms, which include cognitive decline, mood and personality disorders, and loss of motor control[2]. CAG length is roughly correlated inversely with the age of disease onset with repeats longer than ~60 causing a juvenile form of HD[3,4]. No disease-modifying treatments are available. Neuropathologically, degeneration of medium spiny neurons in the striatum and cortical atrophy serve as prominent manifestations[5].

Methods to decipher the pathogenesis of neurodegenerative disorders are needed to identify biomarkers sensitive to clinical progression and to inform therapeutic trials. A potential resource for such insights are patient-derived induced pluripotent stem cells (iPSCs)[6,7], which can be differentiated into multiple cell types, including neurons[8] exhibiting disease phenotypes. Indeed, a number of HD-associated phenotypes have been recapitulated in neurons differentiated from HD iPSCs, including transcriptional dysregulation[4,9–12], bioenergetic deficits[13], impaired neurodevelopment[4,9–12,14,15], altered cell adhesion[10,12],

impaired nucleocytoplasmic trafficking[16,17] and increased susceptibility to cell stressors[18], among others.

The propensity of mHTT to aggregate in neuronal cells is a hallmark of HD and leads to the appearance of large (micrometer scale) nuclear and neuritic inclusions, as seen in mouse models[19] and human post-mortem brain[20]. mHTT's potential role in the impairment of autophagy in HD may contribute to aberrant protein accumulation[21]. However, neither protein aggregation nor disruptions to protein homeostasis have been observed in iPSC-derived HD models unless treated with an inhibitor of the proteasome[22], likely because these models represent early developmental stages where overt disease phenotypes are challenging to detect.

In parallel, other technological advances in cell, molecular and structural biology are poised to contribute to the goal of identifying early markers of pathology. For example, advances in cryogenic electron microscopy (cryoEM) and tomography (cryoET) have recently elucidated the structure of soluble HTT in complex with HAP40 at near-atomic resolution[23], the topology of mHTT-exon 1 and polyQ in vitro aggregates at nanometer resolution[24], and have enabled visualization of the interactions between mHTT-exon 1 aggregates and other proteins and cellular compartments in transfected yeast[25] and HeLa[26] cells, and with molecular chaperones in vitro[27,28]. CryoET and cryoEM have shown aggregates in neurons for other neurodegenerative diseases including poly-GA aggregates in amyotrophic lateral sclerosis[29] and frontotemporal dementia[30]. Correlative light and electron microscopy has also been used to image recruitment of mHTT-exon 1 to cytoplasmic aggregates within the single membrane, vesicle-rich endolysosomal organelles[31].

Herein, we used cryoET to visualize neurites from six human iPSC-derived neurons (iPSC-neuron) cell lines endogenously expressing full-length mHTT with a range of normal and pathogenic CAG repeat lengths (Q18, Q20, Q53, Q66, Q77 and Q109). Using the same methods, we also studied mouse primary embryonic cortical neurons from the BACHD transgenic model[32] expressing the full-length human mHTT, the deltaN17-BACHD model (dN17-BACHD) expressing full-length human mHTT lacking the first 17 amino acids[33] and their littermate wild-type (WT) controls. For all our samples, we examined subcellular organelles previously implicated in HD, namely mitochondria[34] and autophagosomes[35], and found marked changes in morphology as compared to controls. We then coupled these ultrastructural observations with mitochondrial proteomics and identified changes in the levels of several mitochondrial and RNA-binding proteins in HD samples. We also developed an artificial intelligence-based semi-automated 3D segmentation method to quantify changes in mitochondrial granule numbers and sizes.

Guided by ultrastructural and proteomic data, we explored the impact of genetic knockdown (KD) of a SUMO E3 ligase Protein Inhibitor of Activated STAT1 (PIAS1), a protein previously linked to the maintenance of proteostasis and synaptic function in HD[36–38]. PIAS1 reduction abrogated both enlarged mitochondrial granules and aberrant aggregates in autophagic organelles in human HD iPSC-neurons. Following PIAS1 knockdown in the BACHD mouse model, enlarged mitochondrial granules were absent, but aggregates in autophagic organelles remained. These findings are consistent with prior studies showing that PIAS1 knockdown is neuroprotective in HD mice and human iPSC-neurons[21,36].

Our investigations provide a platform with which to structurally evaluate in situ organelle phenotypes in thin regions of intact neurons at nanometer resolution, in the presence and absence of potential therapeutics. The paradigm we propose emphasizes the ability to explore early disease manifestations and mechanisms in neurites of intact patient- and mouse-model-derived neurons, which serves as a proof of concept for the utility of cryoET as a structural readout to assess HD phenotypes and to provide a preclinical evaluation of potential therapies.

## Results

### Huntington's disease patient-derived iPSCs differentiated into mature neurons on electron microscopy grids

Neurons differentiated from human iPSCs with pathological- and normal-length CAG tracts in the *HTT* gene provide a platform to study different HD pathological states within their endogenous genetic context[4]. Here, we developed a robust protocol to differentiate iPSC-neurons with characteristics of medium spiny neurons on electron microscopy (EM) gold grids (Supplementary Fig. 1).

We first differentiated iPSCs to neural progenitors and then adapted our prior maturation protocol[4] to allow the cells to grow directly on EM gold grids, eliminating the Matrigel matrix to minimize background densities, thereby maximizing contrast in cryoET images (Supplementary Fig. 1a). The cells survived, differentiated and matured without Matrigel, producing axons and dendrites, and displaying normal neuronal morphology (Supplementary Fig. 1b, c). Differentiation of cells was also performed at half density (see Methods) to increase the likelihood of obtaining grids with only one cell per grid square, minimizing the potential for overlap between cells while maximizing the number of areas suitable for cryoET imaging (Supplementary Fig. 1c). These modifications allowed iPSCs to differentiate into cells with medium spiny neuron-like characteristics directly on the grids, as validated by DARPP32 and CTIP2 co-staining[12] (Supplementary Fig. 1d). Cells were differentiated for 16 days to progenitors, then plated on EM gold grids for terminal differentiation and maturation. After three more weeks (21 ± 2 days) of terminal differentiation, grids were vitrified by rapid plunging into liquid propane on day ~37–39[39,40] to preserve the neurons in a near-native state without chemical fixative or metal stain.

### CryoET data revealed mitochondria with abnormal cristae and enlarged granules in neurites of HD patient iPSC-neurons

The initial motivation to image intact human HD neurons using cryoET was to determine whether we could directly visualize, at nanometer scale, in situ aggregates of native mHTT (endogenous and untagged) that are not visible using other microscopy methods, as well as their surrounding subcellular components. A single iPSC-neuron is a micrometer-sized cell with a thick cell body and long, thin neurites. Since the electron beam cannot penetrate through the cell body of neurons, we extensively surveyed the structural features in the neurites of our HD cells by recording low magnification 2D images, from which we identified potential regions of interest for subsequent higher-magnification cryoET data collection (Supplementary Fig. 2).

Following an iterative search in many different areas on the cryoEM grids for all HD patient iPSC-derived (Q53, Q66, Q77, and Q109) and control (Q18, Q20) neurons, we failed to detect large cytoplasmic inclusions or aggregates such as those previously observed in in vitro mHTT-exon1 expressing cells[24–28,41,42]. However, this exhaustive examination did detect, in numerous regions in the neurites of all samples, abnormally large and dense, discrete, granular features as well as tangled aggregates, both within double membrane-bound compartments. Some of these compartments showed classic features of mitochondria (Fig. 1), as marked by an outer double membrane and the appearance of inner membrane invaginations forming cristae (Supplementary Movie 1).

The large and dense granular features inside mitochondria were consistently present in tens of tomograms of HD cell lines (Q53, Q66, and Q77), and cristae appeared abnormal in most mitochondria in higher polyQ lines (Q53, Q66, Q77, and Q109) particularly visualized in 3D tomograms (Fig. 1d–f). The Q109 line showed extensive dysmorphology, often having no granules at all, but with highly abnormal cristae (Fig. 1f). Importantly, these aberrant features were absent from mitochondria of control iPSC-neurons (Q18 & Q20), though smaller features, consistent with normal mitochondrial granules, were readily observed (Fig. 1a, b). To facilitate 3D visualization, we used a

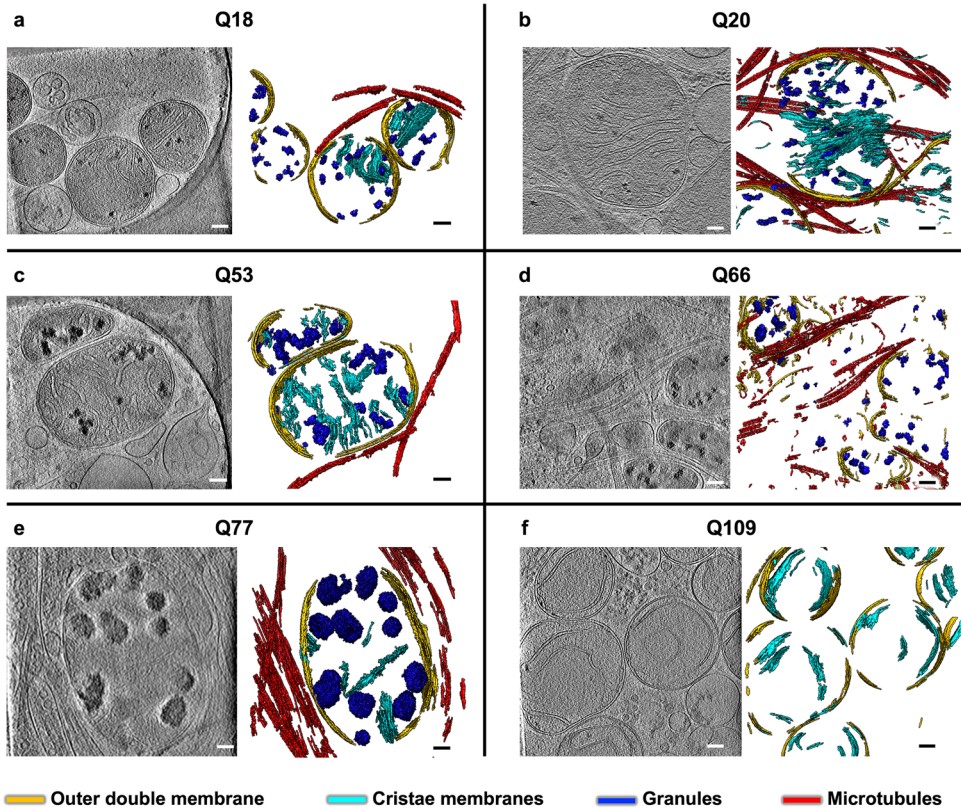

| ⬛ **Outer double membrane** | ⬛ **Cristae membranes** | ⬛ **Granules** | ⬛ **Microtubules** |

**Fig. 1 | Mitochondria in neurites of HD patient iPSC-neurons exhibit altered morphology and contain enlarged granules of varying size.** Slices (-1.4 nm thick) through selected regions of representative cryoET tomograms and corresponding segmentations of local features for **a** Q18, **b** Q20, **c** Q53, **d** Q66, **e** Q77, and **f** Q109 human iPSC-neurons. For HD Qn, mitochondria have swollen cristae and contain enlarged granules compared to controls (Q18 & Q20), with Q109 showing the most aberrant phenotype in which many mitochondria show normal, small granules, some show no granules at all and barely any cristae, and some show large granules. Segmentation colors: red: microtubules, yellow: mitochondrial outer double-membranes, dark blue: granules, and cyan: cristae membranes. Scale bars = 100 nm.

combination of convolutional neural network-based algorithms in EMAN2[43] to annotate and segment these dense structures as well as other mitochondrial and subcellular features in surrounding areas, such as microtubules (Fig. 1a–e, Supplementary Movie 1). Of note, the granules in mitochondria of HD cells did not comprise homogeneous, smooth densities, but rather exhibited complex interwoven textures, occasionally displaying lower-density regions or voids regardless of their size (Fig. 2a, b).

## CryoET data showed mitochondria with abnormal cristae and enlarged granules in neurites of HD mouse model primary neurons

Next, we tested whether the abnormal ultrastructural features we observed in iPSC-neurons were also present in primary embryonic cortical neurons cultured from the BACHD mouse model[32], which expresses full-length human mHTT with an expanded polyQ tract comprised of 97 mixed CAG-CAA repeats under the control of human regulatory sequences. We used cryoET to image neurites in primary cortical BACHD neurons and again found abnormally enlarged granules within mitochondria whose cristae were often disrupted compared to WT (Fig. 3a, b), similar to those seen in neurites of iPSC-neurons (Fig. 1c–f). Our data are evidence that neurons from both human and mouse HD models share the presence of enlarged granules and other changes in mitochondria, supporting the view that these morphological abnormalities could be used as diagnostic features of HD. We also evaluated primary cortical neurons from BACHD transgenic mice expressing human mHTT lacking the first 17 N-terminal amino acids (dN17-BACHD)[33]. The N-terminus of HTT contains a putative mitochondrial membrane-targeting sequence

that can form an amphipathic helix[44,45] characteristic of proteins transported into the mitochondria. Further, this domain is required for interaction with the translocase subunit of mitochondrial inner membrane protein 23 (Tim23), essential for protein import across mitochondria[46]. More severe phenotypes are observed in dN17-BACHD than BACHD mice[33], with nuclear and endoplasmic reticulum (ER) mHTT localization for the former, as well as large nuclear inclusions in the BACHD model[33]. CryoET of neurites in dN17-BACHD neurons showed even more severe distortions in most of their mitochondria (Fig. 3c), suggesting a role for the N17 domain in mitochondrial integrity. These observations, therefore, suggest that the polyQ tract of mHTT can be particularly disruptive to mitochondria, as previously reported for mitochondrial membranes using other techniques[47].

## Neurites in HD patient iPSC-derived and mouse model neurons contain sheet-like aggregates in autophagic organelles

In addition to the dense, enlarged granules observed in mitochondria, we observed numerous, much larger aggregates in other membrane-bound compartments in both HD patient iPSC-neurons and mouse primary neurons. While they appeared to be filamentous when viewed in two-dimensional (2D) z-slices through the tomograms, closer three-dimensional (3D) inspection, revealed that they are composed of densely interwoven slab-shaped, and sheet-like aggregates (Fig. 4, Supplementary Movie 2), hereafter called sheet aggregates.

Importantly, these sheet aggregates were present in the neurites of all HD patient Qn neurons (Q53, Q66, Q77, and Q109) (Fig. 5a) as well as in those of HD mouse model neurons (BACHD and deltaN17-BACHD) (Fig. 5b). On the other hand, double membrane-bound

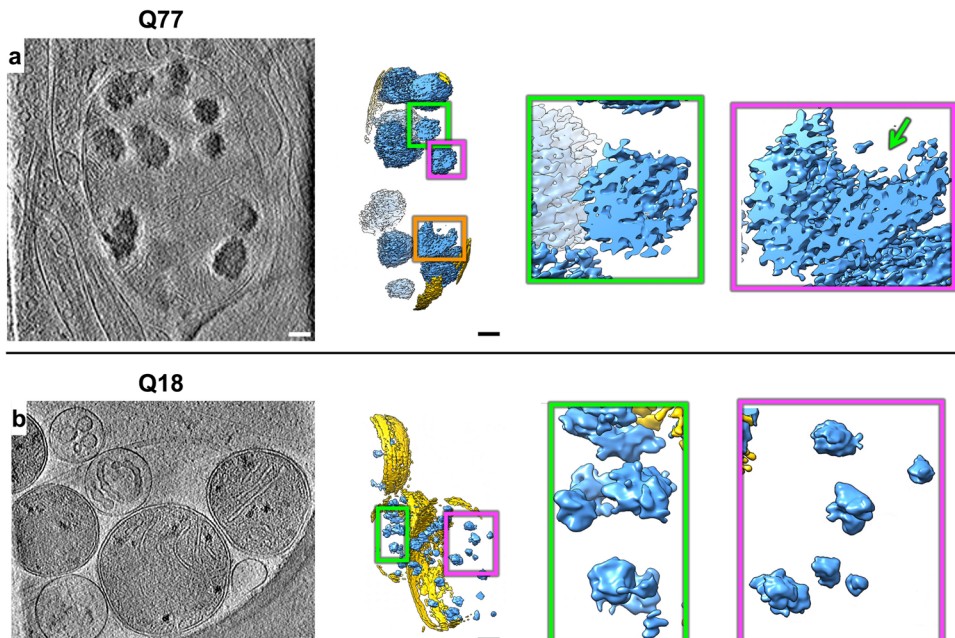

**Fig. 2 | Mitochondria in HD neurites contain enlarged granules composed of tightly packed, heterogeneous densities.** Z-slices (-1.4 nm thick) through representative cryoET tomograms of a neurite from **a** a HD iPSC-neuron (Q77) and **b** a control iPSC-neuron (Q18), and corresponding oblique cutaway-view segmentations of mitochondrial outer double membranes (yellow) and dense, granular densities inside (light blue), with accompanying zoomed-in views of the mitochondrial granules. Scale bars = 100 nm. Segmentation colors: yellow: double membrane, light blue: mitochondrial granules.

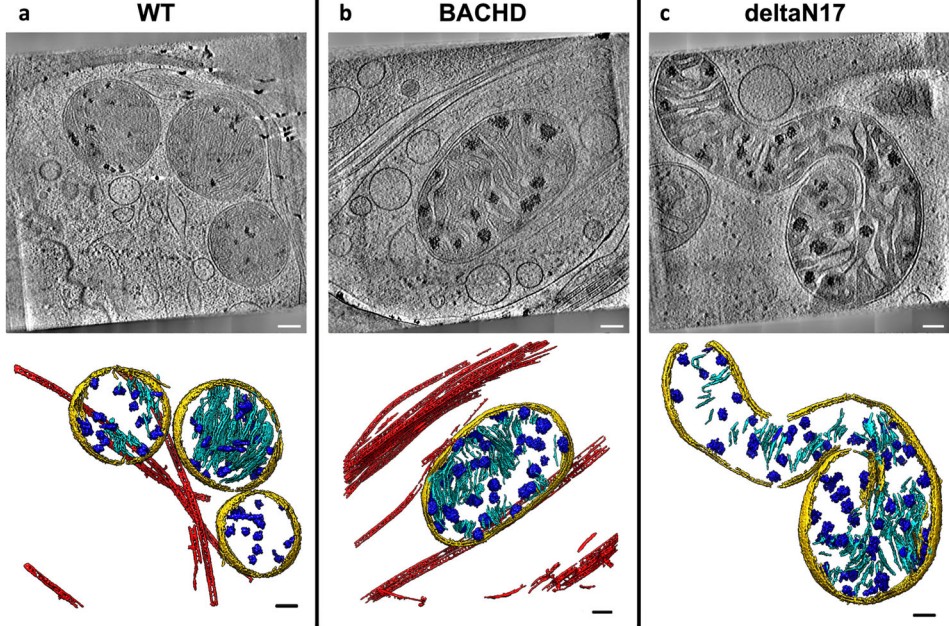

**Fig. 3 | Mitochondria from neurites of HD mouse model neurons exhibit altered morphology and contain enlarged granules of varying size.** Slices (-1.4 nm thick) through selected regions of representative cryoET tomograms and corresponding segmentations of local features for **a** WT, **b** BACHD and **c** dN17 BACHD primary neurons reveal that neuronal mitochondria in HD mice have swollen cristae and contain enlarged granules compared to controls (WT). Segmentation colors are the same as Fig.1. Scale bars = 100 nm.

compartments in neurites of control human iPSC (Q18) (Fig. 5c) and mouse (WT) (Fig. 5d) neurons lacked these sheet aggregates.

Of note, the sheets in these aggregates were reminiscent of some regions in mHTT exon 1 and polyQ-only aggregates in vitro, shown to contain long, relatively flat, or slightly curved, ribbon-like[48] or sheet-like[24] morphologies. Similar features were also observed previously for poly-GA aggregates[29] and possibly in small regions of GFP-tagged TDP-25 gel-like inclusions[30]. Interestingly, the thickness of the sheet aggregates in our neurons here was more uniformly ~2 nm when visualized in our 3D tomograms with contrast transfer function correction and without downsampling or low-pass filtration. Fourier transforms of these sheets in higher magnification images did not reveal any periodic arrangement in them.

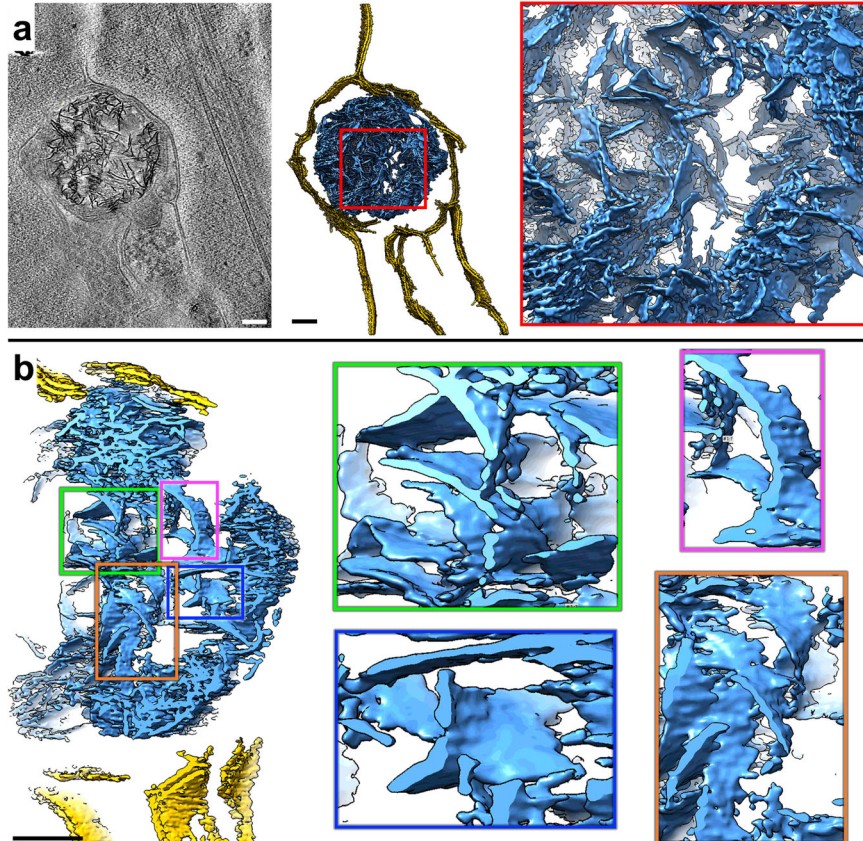

**Fig. 4 | Neurites in HD cells contain double membrane-bound compartments with sheet aggregates composed of interwoven slabs and sheets. a** Z-slice (~1.4 nm thick) through a selected region showing a sheet aggregate in a representative cryoET tomogram of a neurite of an HD patient iPSC-neuron (Q66) and corresponding segmentation of double membranes (yellow) and aggregated densities inside (light blue). **b** Cutaway, oblique view of an enlarged region from the segmentation in **a** and further zoomed-in views of selected subregions showing examples of sheet-like areas within the aggregate. Scale bars = 100 nm. Segmentation colors: yellow: double membrane, light blue: sheet aggregate.

Eukaryotic cells contain several characteristic double membrane-bound compartments including mitochondria, nucleus, and organelles in the autophagy pathway such as mitochondria-derived vesicles[49], autophagosomes[50], and amphisomes[50]. Here, the compartments containing the sheet aggregates (~200 nm to ~500 nm range in the longest span) were much smaller than the nucleus (~3–18 μm in diameter), and sometimes seemed to bind or merge with one another (Supplementary Fig. 3a). While they were most often similar in size to small mitochondria, and could possibly correspond to degenerating versions of this organelle or mitochondria-derived vesicles[49], they may also correspond to other autophagic organelles with no visible cristae or to other molecular components targeted for autophagy[50], suggesting alternative and possibly complementary or parallel biogenesis origins. For instance, autophagosomes participate in cellular pathways for degradation and clearance and thus are possible candidates to contain these sheet aggregates. Supporting this interpretation, we found instances of sheet aggregates within double membrane-bound compartments fused with single membrane-bound compartments (Supplementary Fig. 3b–d), reminiscent of lysosomes, a picture strikingly similar to amphisomes, which result from the fusion of autophagosomes with lysosomes[50].

In addition, in support of the degenerating mitochondria assignment, we observed features suggestive of nascent sheet aggregates in what appeared to be degenerating mitochondria with disrupted cristae remnants (Supplementary Fig. 3e). To further characterize the nature of these features, we again used semi-automated, neural-networks-based annotation of the corresponding tomogram with EMAN2[43], training on a few positive references (*n* = 10) from a mature sheet aggregate only. Strikingly, the algorithm assigned the putative nascent sheet aggregate features in what appears to be a mitochondrion as belonging to the same type of feature as mature sheet aggregates, even though the networks were trained exclusively with mature sheet aggregate references (Supplementary Fig. 3f). This organelle could represent a degenerated mitochondrion associated with a lysosome, as these two organelles were recently discovered to interact directly via their membranes[51]. Indeed, this mitochondrion and a similar neighboring organelle are seen interacting with single membrane-bound compartments (Supplementary Fig. 3g).

### Mitochondrial proteomics of human iPSC-neurons identified differentially expressed proteins, including those engaged in RNA binding

The accumulation of mitochondrial granules (Fig. 2), distinctly different from the sheet aggregates (Fig. 4), and the disruption of cristae observed in the neurites of HD neurons, are consistent with impaired mitochondrial function and bioenergetics previously described for HD[34,52]. To investigate potential mechanisms underlying the abnormal enlargement of mitochondrial granules, we performed liquid chromatography-tandem mass spectrometry-based proteomic analysis on mitochondria isolated from HD patient iPSC-neurons (Q109) and controls (Q18) since the former represents the most extreme phenotype among our HD patient iPSC-neuron samples (Fig. 6). The Q109 line has severely disrupted mitochondria with few granules and highly disrupted cristae. Mitochondria were isolated by using magnetically labeled anti-TOM22 microbeads[53].

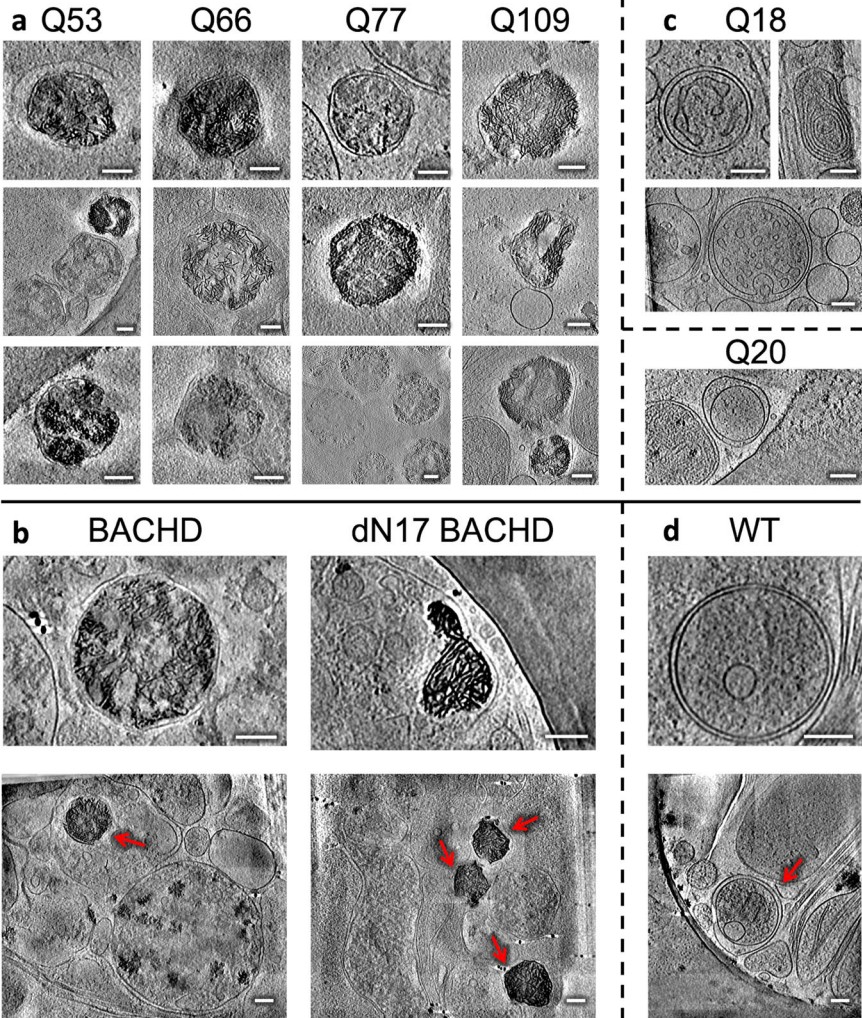

**Fig. 5 | Neurites in HD patient iPSC-derived and mouse model primary neurons exhibit sheet aggregates within double membrane-bound organelles.** Slices (-1.4 nm thick) through selected regions of representative cryoET tomograms showing double membrane-bound compartments in neurites of **a** iPSC-neurons and **b** mouse model primary neurons, as well as **c** Q18, Q20, and **d** WT controls. Control lines do not show the presence of sheet aggregates. Scale bars = 100 nm. The red arrows in full-frame views in **b** and **d** indicate double membrane-bound organelles.

CryoET tomograms of isolated mitochondria (Fig. 6a, b) looked like those in HD neurites, containing enlarged granules in Q109 and small granules in Q18 (Fig. 1a, f, Fig. 6a, b). In quality control experiments, the mitochondrial isolation method was assessed via Western analysis on each cell fraction; this yielded a fraction with enrichment of mitochondria (using ATPB protein levels as a proxy) with some minor contamination from other organelles (LC3 levels were used as a proxy for autophagosomes, and CTIP2 levels as a proxy for nuclei) (Supplementary Fig. 4a). Consistent with previous observations, our proteomic datasets were associated with Gene Ontology (GO) terms related to mitochondrial functions (Supplementary Fig. 4b).

Comparing the mitochondrial proteome of Q109 HD patient iPSC-neurons vs controls (Q18) revealed a total of 177 differentially enriched peptides, of which 124 were unique proteins, the majority of which were depleted in HD (Fig. 6c & Supplementary Data 1). Mitochondrial-encoded proteins were not depleted, however, depleted proteins included nuclear-encoded mitochondrial proteins (purple in Fig. 6d) such as TOMM70A, a mitochondrial import receptor involved in the translocation of preproteins which contain a mitochondrial targeting sequence, into mitochondria, which is impaired in HD[47]. Interestingly, FIS1 (mitochondrial fission protein) levels were increased in HD (Fig. 6d), consistent with previous data showing altered regulation of mitochondrial fission in HD[54,55].

GO analysis of the differentially enriched peptides found RNA binding to be the most significantly altered molecular function (Fig. 7a), as reflected by increased levels of various cytoplasmic RNA binding proteins such as hnRNPA2B1, hnRNPA1, and hnRNPH1. G-rich sequence factor 1 (GRSF1), a nuclear-encoded mitochondrial RNA binding protein essential for mitochondrial homeostasis and required for mitochondrial RNA processing[56,57], showed reduced levels in the Q109 HD mitochondria, consistent with the general lack of granules in these neurons. Panther pathway analysis identified cytoskeletal regulation by Rho GTPase and axon guidance mediated by semaphorins as the two most overrepresented pathways by the DEPs (Fig. 7b).

Ingenuity Pathway Analysis (IPA) of the mitochondrial differentially enriched peptides identified a network representing proteins involved in RNA post-transcriptional modification, further implicating these proteins in RNA biology and dynamics (Fig. 7c). Upstream regulators identified by IPA included Amyloid Precursor Protein (APP) and transforming growth factor β (TGF-β) as predicted inhibitors of the mitochondrial DEPs in HD. IPA pathways included mitochondrial dysfunction and Glycolysis 1, consistent with the mitochondrial deficits in HD[58,59] (Supplementary Fig. 4c–f, Supplementary Data 1).

The presence of enlarged mitochondrial granules could represent aberrant accumulation of mitochondrial RNA granules (MRGs)[56,60,61], suggesting RNA processing and RNA quality control

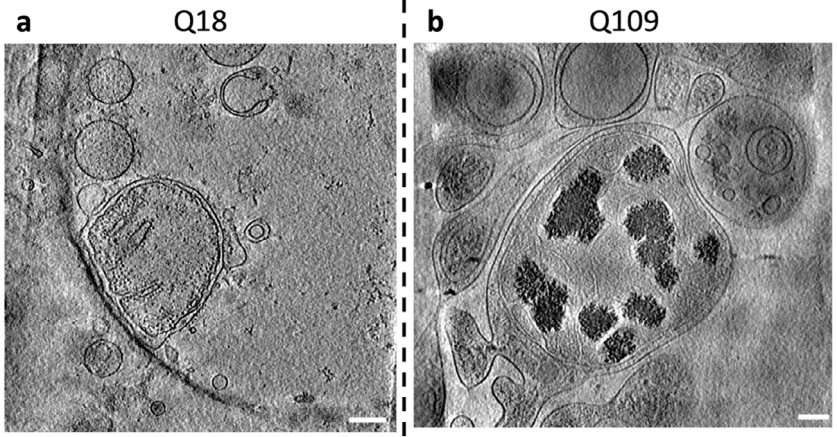

**c** No. of unique differentially enriched proteins in mitochondria from 109Q vs 18Q neurons

| Depleted, 97 | Enriched, 27 |

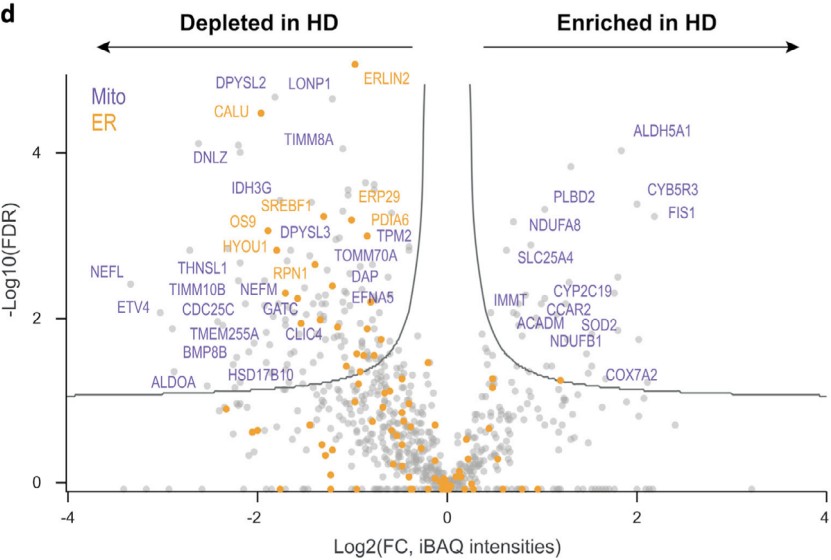

**Fig. 6 | Mass spectrometry of isolated mitochondria revealed different levels of associated proteins in neurites of iPSC-neurons (Q109) vs controls with representative cryoET of isolated HD mitochondria.** Z-slices (-1.4 nm thick) of **a** control Q18 and **b** Q109 mitochondria showing abnormal accumulation and enlargement of mitochondrial granules in the latter. Scale bar = 100 nm. Most of the mitochondria from Q109 have aberrant cristae and no granules as shown in Fig. 1 and quantitated in Fig. 9. **c** Mass spectrometry of proteins in isolated mitochondria showed 177 differentially enriched peptides, of which 124 were unique to a certain protein. **d** Scatter plot showing fold change of peptides(X) vs the log false discovery rate(Y) of the 177 peptides highlighting selected proteins that were depleted or increased in HD (Q109) mitochondria in comparison to controls (Q18). Mitochondrial proteins are colored in purple and ER proteins in orange.

deficits, and/or aberrant protein quality control due to disrupted protein import arising from the presence of mHTT[46]. Given that mass spec shows a differential enrichment of RBPs, which are components of MRGs, we investigated whether the enlarged granules might be MRGs by modulating GRSF1, which is involved in the formation of MRGs[56,57], in the Q66 neurons that showed significantly and consistently enlarged mitochondrial granules (Fig. 1d) but less disruption of mitochondrial cristae. GRSF1 was reduced through RNAi-mediated knockdown and we examined whether there was any effect on granule size and/or number per mitochondria. Accell (Horizon Discovery) siRNA treatment was initiated on day 28 of differentiation after plating on grids at day 16. Total knockdown at day 37 was recorded by qRT-PCR at 90% (Supplementary Fig. 4g–i). Tomograms were collected from the cells on grids. The enlarged granules were disrupted (Fig. 7d), and were of significantly smaller size as reflected in subsequent quantitative analysis (described below), following GRSF1 knockdown compared to control scrambled siRNA, suggesting the granules are MRGs.

## *PIAS1* heterozygous knockout in HD patient iPSC-neurons and short-term *Pias1* knockdown in BACHD mouse neurons modulate aberrant organelle phenotypes

Mitochondrial dysfunction can be highly detrimental to neuronal function, particularly in light of the extensive energetic requirements for synaptic function[62]. To evaluate whether the observed phenotypes can be ameliorated, we evaluated genetic reduction of an E3 SUMO ligase, PIAS1, based on our previous data showing that reduced *Pias1* expression resulted in: decreased aberrant accumulation of mHTT in R6/2 HD mice, rescued transcriptional deficits in zQ175 HD mice, particularly of synaptic genes, and improved mitochondrial DNA integrity and synaptic gene expression in iPSC-neurons[36,38]. PIAS1 enhances SUMOylation of various proteins, including HTT[37,63]. To

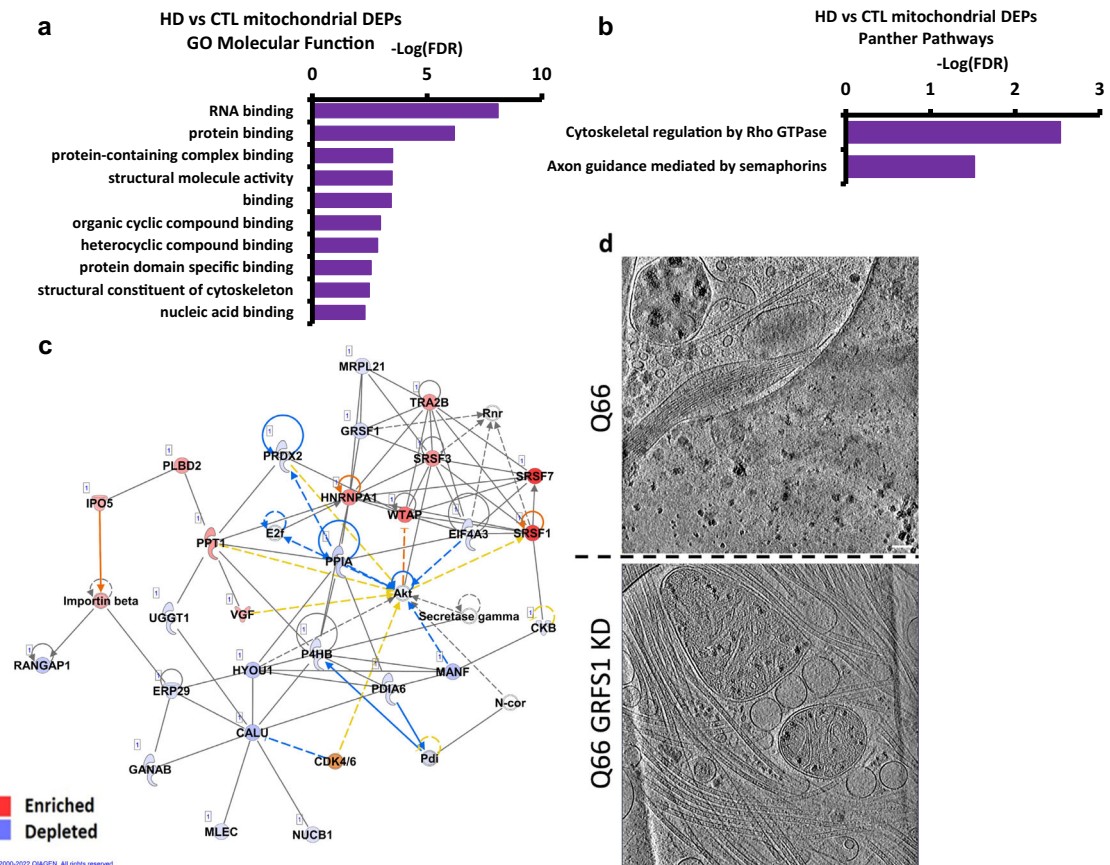

**Fig. 7 | Analysis of mass spectrometry data from isolated HD (Q109) vs control mitochondria indicates RNA binding and knockdown of RNA binding protein *GRSF1* in Q66 reduces mitochondria granule size. a** Graph of gene ontology (GO) analysis of the 124 unique of differentially enriched proteins in HD vs control mitochondria showing molecular functions overrepresented by HD DEPs. The false Discovery Rate (FDR) indicates the reliability of the functional identification of the DEPs. **b** Graph of Panther analysis of the 124 differentially enriched proteins in HD vs control mitochondria showing Panther Pathways overrepresented by HD

mitochondria differentially enriched proteins. **c** Ingenuity pathway analysis of the differentially enriched proteins between HD and control mitochondria highlighted Post-Translational Modification, RNA Post-Transcriptional Modification, and Protein Folding as the top network (score, 60 Focus molecules: 27). Proteins in blue are depleted in HD while proteins in red are enriched. **d** Examples of z-slices (-1.4 nm thick) through cryoET tomograms of day 37 iPSC-neurons (Q66) without (top) and with (bottom) treatment at day 28 with *GRSF1* siRNA. Scale bars = 100 nm.

computationally determine whether targeting PIAS1 would predict changes in the HD mitochondrial proteome, we first compared the changes in the mitochondrial proteome of HD iPSC-neurons (Q109 vs Q18) described above with findings from our prior study[36] of gene expression in the same type of differentiated neurons following PIAS1 knockdown (Supplementary Fig. 5a, b). We found a significant overlap between mitochondrial differentially enriched peptides and RNA changes induced by siRNA depletion of PIAS1 in HD iPSC-neurons (representation factor:2.1 $p < 3.314$e-05, Supplementary Fig. 5a, b, Supplementary Data 1). These comparisons further supported investigating whether knockdown of PIAS1 could influence the presence and/or size of the aberrant granules we observed within HD mitochondria.

We again used cryoET to visualize iPSC-neurons (Q53 & Q66, representing an intermediate range of phenotypes) with *PIAS1* heterozygous knockout (hetKO), using a CRISPR-Cas9-generated hetKO, which produces approximately 50% knockdown (Supplementary Fig. 6a–d). The *PIAS1* hetKO iPSC neurons differentiated well on EM gold grids in preparation for cryoET experiments (Supplementary Fig. 6e). The Q53 and Q66 *PIAS1* hetKO tomograms (Fig. 8a, b **right panels**) showed seemingly healthy mitochondria and other double membrane-bound organelles, lacking the abnormally enlarged granules and sheet aggregates, respectively, as had been observed in HD patient iPSC-neurons for all Qns (Figs. 1, 5 & 8a, b **left panels**). Indeed,

the structural features of Q53 and Q66 *PIAS1* hetKO neurites resemble those from control Q18 neurons rather than the parental HD lines.

To determine if the rescue of abnormal morphologies observed in HD patient iPSC-neurons (Q53 & Q66) with *PIAS1* hetKO translates to mouse cortical neurons, we carried out a short-term *Pias1* KD in mouse primary neuronal cultures derived from E18 BACHD cortical neurons on EM grids and visualized cells with cryoET (Fig. 8c, d). To reduce *Pias1* levels, Accell (Dharmacon) siRNA smart pools against mouse *Pias1* were used. *Pias1* KD was initiated at day in vitro 3 (DIV3) with one treatment and grown for 11 days. Cells were vitrified for cryoET analysis at DIV14. Knockdown of *Pias1* in the BACHD neurons was successful according to qRT-PCR analyses; 43% knockdown was achieved comparing control-siRNA-treated neurons to *Pias1* knockdown (Supplementary Fig. 6g, h).

CryoET experiments showed that treating BACHD neurons with *Pias1* siRNA resulted in a reduction of enlarged granules in neurons, with many mitochondria completely lacking detectable granules in comparison to the BACHD mitochondria. However, sheet aggregates in autophagic organelles were present in comparable numbers to those in BACHD neurons without *Pias1* KD (Fig. 8d, bottom right). Thus, the beneficial effects of *PIAS1* hetKO in HD patient iPSC-neurons was only partially replicated in the mouse model under our experimental conditions, possibly due to the later and shorter treatment time frame. Whether the impact of PIAS1 reduction is consistent across

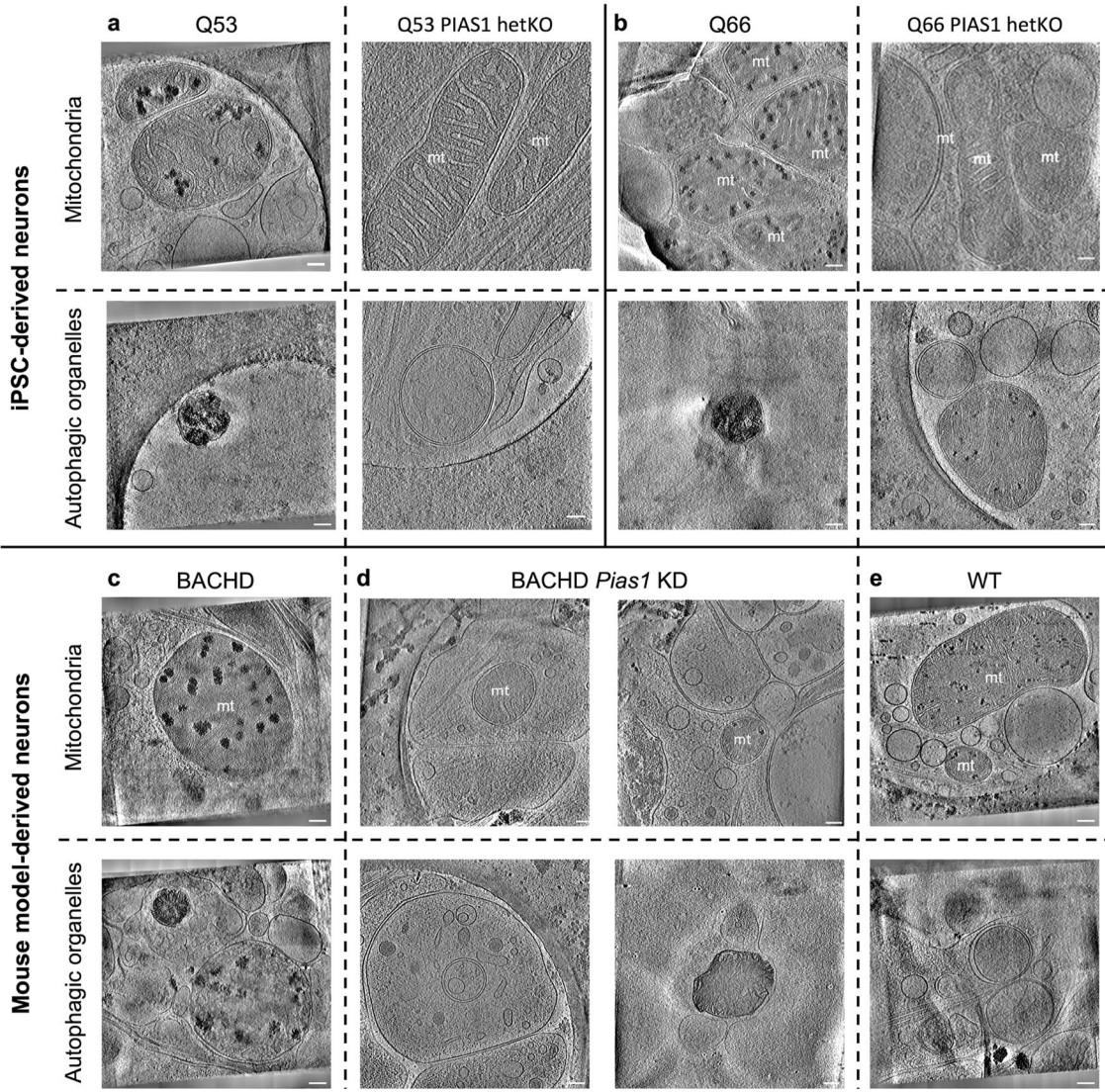

**Fig. 8 | Effects of reduced *PIAS1/Pias1* on mitochondrial granule size and sheet aggregates in autophagic organelles in neurites of HD model neurons.** Slices (~1.4 nm thick) through representative cryoET tomograms of neurites in **a** Q53 and **b** Q66 HD iPSC-neurons without (left) and with (right) *PIAS1* hetKO show that *PIAS1* hetKO ameliorated the phenotypes of enlarged mitochondrial granules (first row) and sheet aggregates in putative autophagic organelles (second row). Slices (~1.4 nm thick) through representative cryoET tomograms of neurites in **c** BACHD, **d** BACHD *Pias1* KD, and **e** WT mouse neurons show that *Pias1* KD ameliorated the enlarged mitochondrial granules (third row) but not the sheet aggregates in putative autophagic organelles (fourth row) in BACHD-derived neurons. Scale bars = 100 nm.

all cells or results in different effects in different cell types needs further investigation.

### Artificial intelligence-based semi-automated 3D segmentation enabled quantitative characterization of abnormally enlarged mitochondrial granules in neurites of HD neurons

We observed both enlarged granules and disrupted cristae in the mitochondria of HD neuronal processes (Figs. 1–3). The mitochondrial granules varied in size, seeming much larger in HD mitochondria than in controls. To quantify their size distribution, we developed a semi-supervised, artificial intelligence-based method to automatically detect and segment mitochondria and mitochondrial granules in neurite tomograms (Supplementary Fig. 7). Among all tomograms gathered from three technical replicates per cell line (Supplementary Table 1), the algorithm found that 139 contained mitochondria for the various HD patient iPSC-neurons and 83 for mouse model neurons (Supplementary Table 1).

Quantifying mitochondrial granule volumes by segmentation estimation (see Methods) (Fig. 9) with our algorithm yielded distributions for Q53, Q66, and Q77 that were shifted towards larger sizes with respect to controls (Fig. 9a). This was also observed for BACHD and, dN17-BACHD compared to WT neurons (Fig. 9c), transfected with a control siRNA. In contrast, granule volumes on average were not larger in Q109 compared to controls, likely due to the significant general disruption of mitochondria in the former. PIAS1 hetKO produced a significant decrease in the Q66 and Q53 lines when compared to parental Q66 and Q53 (Dunn's multiple comparisons Q66 vs. Q66 PIAS1 hetKO padj < 0.0001, Q53 vs. Q53 PIAS1 hetKO padj < 0.0001), with Q66 PIAS1 hetKO resembling the control Q18 and Q20 iPSC lines, and Q53 PIAS1 hetKO having granules significantly smaller than controls (Q20 vs. Q53 PIAS1 hetKO padj < 0.0001 Q18 vs. Q53 PIAS1 hetKO padj = 0.0002).

Furthermore, mitochondrial granule volumes were rescued to WT size in BACHD neurons treated with *Pias1* siRNA in comparison to BACHD neurons without treatment and BACHD neurons treated with

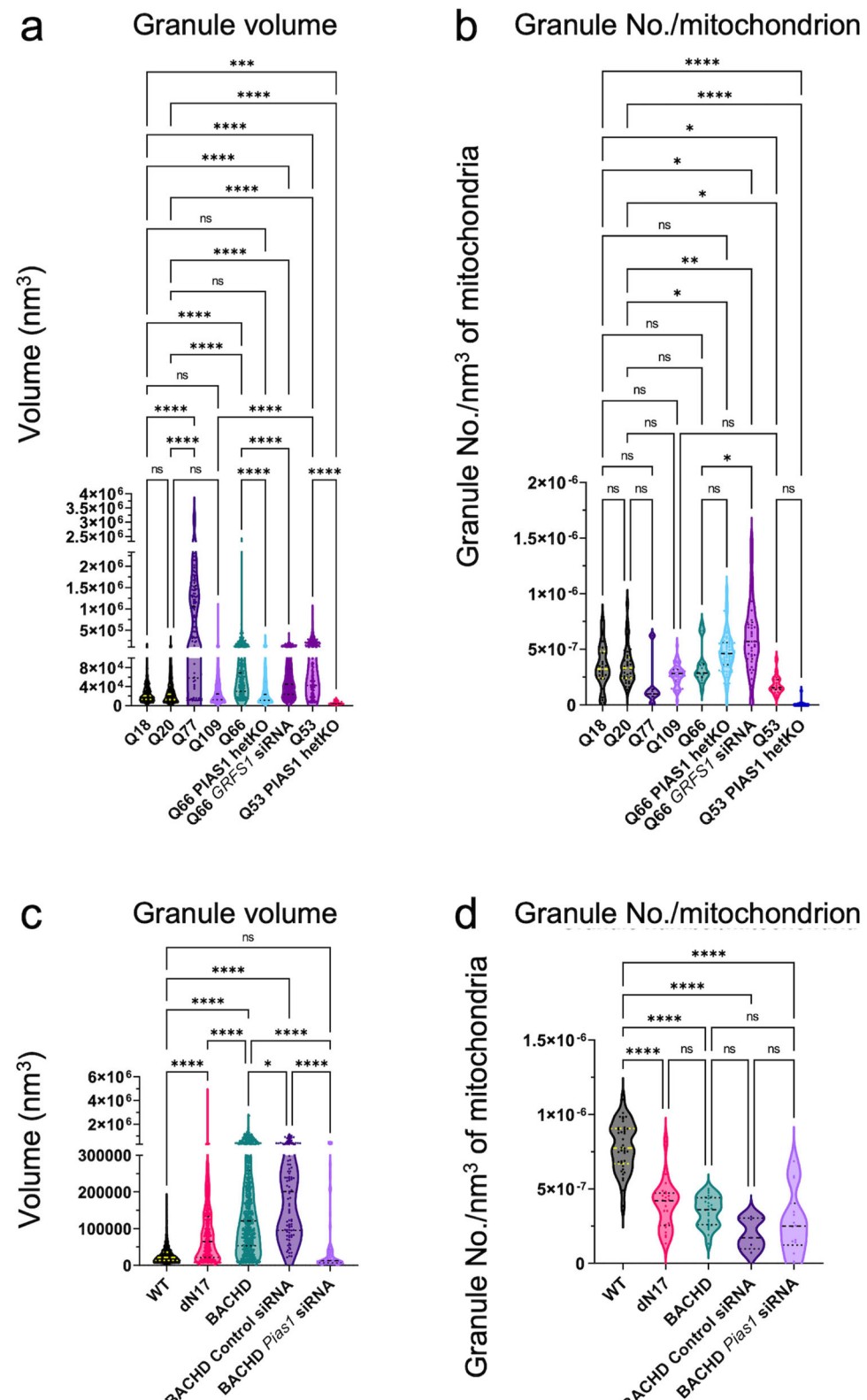

control siRNA (Dunn's multiple comparisons WT vs BACHD *Pias1* siRNA padj > 0.9999, BACHD vs. BACHD *Pias1* siRNA padj < 0.0001, and BACHD Control siRNA vs. BACHD Pias1 siRNA padj < 0.0001). In addition, there was a significant decrease in granule volume when we treated Q66 neurons with *GRSF1* siRNA (padj < 0.0001).

While mitochondrial granule volumes were increased for all Qn iPSC-neurons except Q109, granule numbers per nm$^3$ of

mitochondrial volume were not different in any of them from those in control Q18 or Q20 cells (Fig. 9b). Interestingly, *GRSF1* siRNA treatment of Q66 neurons significantly increased the number of granules in the mitochondria (padj = 0.0316). On the other hand, granule numbers per nm$^3$ of mitochondrial volume were decreased in all mouse cell lines with respect to WT, including in those treated with siRNAs (Fig. 9d).

**Fig. 9 | Artificial Intelligence (AI)-based 3D quantification of mitochondrial granule volume and granule number per mitochondrial volume.** AI-based quantitation of neuronal tomograms containing mitochondria demonstrates a significant increase in mitochondrial granule size with higher polyQ in HD human and mouse model neurons. While this is rescued by *PIAS1* hetKO and *GRSF1* siRNA in iPSC neurons, it is only partially rescued by *Pias1* KD in mouse model neurons. Violin plots displaying AI measurements of mitochondrial granule **a** volume (Kruskal Wallis statistic = 857.2, *P* value < 0.0001 with Dunn's multiple comparisons) and **b** numbers per nm³ of mitochondrial volume (Kruskal Wallis statistic = 129.6 *P* < 0.0001 with Dunn's multiple comparisons) from cryoET tomograms of neurites for several HD patient iPSC-neuronal cell lines (number of tomograms per cell line: Q18 = 21, Q20 = 20, Q53 = 14, Q66 = 10, Q77 = 5, Q109 = 37, Q53 P*IAS1* hetKO = 22, Q66 *PIAS1* hetKO = 68 and Q66 *GRSF1* siRNA = 17; number of granules per cell line:Q18 = 364, Q20 = 774 Q53 = 254, Q66 = 249, Q77 = 94, Q109 = 1534, Q53

P*IAS1* hetKO = 13, and Q66 *PIAS1* hetKO = 2524 and Q66 *GRSF1* siRNA = 958) as well as mitochondrial granule **c** volume (Kruskal Wallis statistic = 1030, *P* < 0.0001 with Dunn's multiple comparisons) and **d** numbers per nm³ of mitochondrial volume (Kruskal Wallis statistic = 68.28, *P* < 0.0001 with Dunn's multiple comparisons) from cryoET tomograms of neurites for three mouse neuron models (number of tomograms per mouse model: WT = 31, BACHD = 22, dN17 BACHD = 15, BACHD Control siRNA=5 and BACHD *Pias1* siRNA=12; number of granules per mouse model:WT = 2057, BACHD = 634, dN17-BACHD = 477, BACHD Control siRNA = 91 and BACHD *Pias1* siRNA = 92). The human neurons and mouse model neurons showed an increase in granule volumes in all but Q109, a trend of reduced granule number in human neurons and a significant reduction in granule number in mouse model neurons. Median displayed as dashed line, dotted lines display quartiles. ns = not significant, ****p < 0.0001, **p < 0.01, *p < 0.05 For full statistical details refer to Supplementary Table 2. Source data are provided as a Source Data file.

## Discussion

In this study, we have defined Q-length dependent ultrastructural changes that occur in neuronal processes (neurites) of human HD iPSC-neurons and BACHD primary cortical neurons. Specifically, we found that HD neurons contain two types of double membrane-bound organelles including mitochondria and autophagic organelles with abnormal densities inside, which are completely absent in healthy control neurons. First, in mitochondria, we observed ultrastructural changes, most notably enlarged granules in all HD samples compared to controls (Figs. 1–3). Importantly, many HD samples also exhibited severely disrupted cristae, similar to cryoET observations in other neurodegenerative disorders such as Leigh syndrome[64,65]. Second, we observed sheet aggregates within autophagic organelles resembling mitochondria-derived vesicles, autophagosomes and/or amphisomes (Figs. 4, 5, Supplementary Fig. 3). These findings are highly significant in demonstrating the disruption of organellar structure in HD, possibly as very early events in pathogenesis that precede overt neuronal dysfunction and the appearance of inclusions visible in neurons derived from HD patient[20] and mouse model[19,66] brain tissues.

Cellular cryoET experiments yield 3D volumes ("tomograms") that sample regions of the crowded subcellular environment in vitrified cells preserving their native structures. In the initial stages of this project, we visualized hundreds of neurite tomograms, leading to the discovery of enlarged granules in mitochondria and sheet aggregates in autophagic organelles within them. Using a newly developed semi-supervised artificial intelligence-based method to segment and quantify the number and volume of mitochondrial granules (Supplementary Fig. 7), we concluded that their enlargement is a structural signature of HD, consistently present in both human iPSC- and mouse model-derived neurons (Fig. 9). Figure 9 shows the statistical analysis of all our tomograms with mitochondria from various HD models and treatments. Thus, our quantitative findings do not depend on subjective observation.

The aberrant accumulation of large mitochondrial granules and abnormal cristae are known hallmarks of mitochondrial dysfunction as assayed by other methods in similar systems[34,52]. As members of the HD iPSC Consortium[10], we previously showed that striatal-like HD iPSC-neurons similar to those examined here have mitochondrial deficits including altered mitochondrial oxidative phosphorylation and enhanced glycolysis[13]. Additional studies have also shown mitochondrial dysfunction, fragmentation, and disrupted cristae by electron microscopy of chemically fixed cell lines expressing mHTT[67].

The high scattering contrast of the enlarged granules we observed here in both human and mouse HD model neurons could be attributable to RNA and/or calcium phosphate. Mitochondria normally contain calcium phosphate granules, which store excess calcium, maintain mitochondrial calcium concentration, and contribute to the maintenance of mitochondrial function[68,69]. Furthermore, calcium overload within cells causes ultrastructural remodeling of cristae[70,71], as in our experimental results described above, which was particularly

extensive for Q109 iPSC-neurons and dN17-BACHD neurons. Cellular calcium dyshomeostasis is a well-established phenotype in HD[72–74] and mitochondrial calcium dysregulation was observed in mitochondria isolated from transgenic YAC128 HD mice[75]. Assessing the proteome of mitochondria isolated from human HD iPSC-neurons identified differentially enriched proteins related to protein import and RNA binding (Fig. 7). Mitochondrial RNA granules (MRGs) are normal features of the mitochondrial matrix[57], and are composed of newly synthesized RNA, RNA processing proteins and mitoribosome assembly factors[57,61,76–78]. Stressors such as dysregulation of RNA processing and RNA quality control defects or inhibition of mitochondrial fission or fusion[60,61] can cause aberrant accumulation of MRGs, which are composed of large ribonucleoprotein complexes[60]. MRGs are fluid condensates[61] and components of mitochondrial post-transcriptional pathways and are responsible for mitochondrial RNA translation[79]. When these granules aberrantly accumulate, the integrity of cristae is compromised to accommodate them[61]. The enrichment of RNA binding proteins in the proteome of the Q109 HD iPSC-neurons versus controls suggested that these enlarged granules may represent MRGs. GRSF1 is a nuclear-encoded RNA-binding protein that regulates RNA processing in MRGs[56] and is critical for maintaining mitochondrial function[70,80]. GRSF1 has also been implicated in cellular senescence with levels declining in senescent cells and lowered GRSF1 levels causing mitochondrial stress[70]. Following knockdown of *GRSF1* in the Q66 HD neurons, which show significantly and consistently enlarged mitochondrial granules, GRSF1 reduction led to a decrease in granule size and disbursement of smaller granules (Fig. 9a) consistent with the granules representing MRGs. While the granules are likely MRGs, there remains the possibility that calcium homeostasis is also disrupted, contributing to HD mitochondrial phenotypes leading to the formation of calcium phosphate as one of the constituents in the observed granules. Future studies such as elemental analysis with electron microscopy[81] will help further define the chemical nature of these granules.

We observed differences in the mitochondrial phenotype between the mouse models. While both BACHD and dN17-BACHD mice showed enlarged mitochondrial granules with respect to controls (albeit fewer in number), dN17-BACHD displayed smaller granules than BACHD and severely distorted structures, such as seemingly enlarged mitochondria and swollen cristae. Our data provide additional hints regarding potential mechanistic underpinnings of normal HTT function including protein turnover and mitochondrial protein import. In HD, mitochondrial function appears to be impacted by the altered import of mitochondrial precursor proteins in the presence of mutant HTT[46]. Indeed, the high-affinity interaction of mHTT with the inner mitochondrial protein import complex subunit TIM23 disrupts the HD mitochondrial proteome[47]. Furthermore, protein aggregation and alterations in RNA processing and quality control can disrupt protein import[82]. More recently, defects in mitochondrial protein import have also been connected to impaired proteostasis[83]. Mutant HTT can also

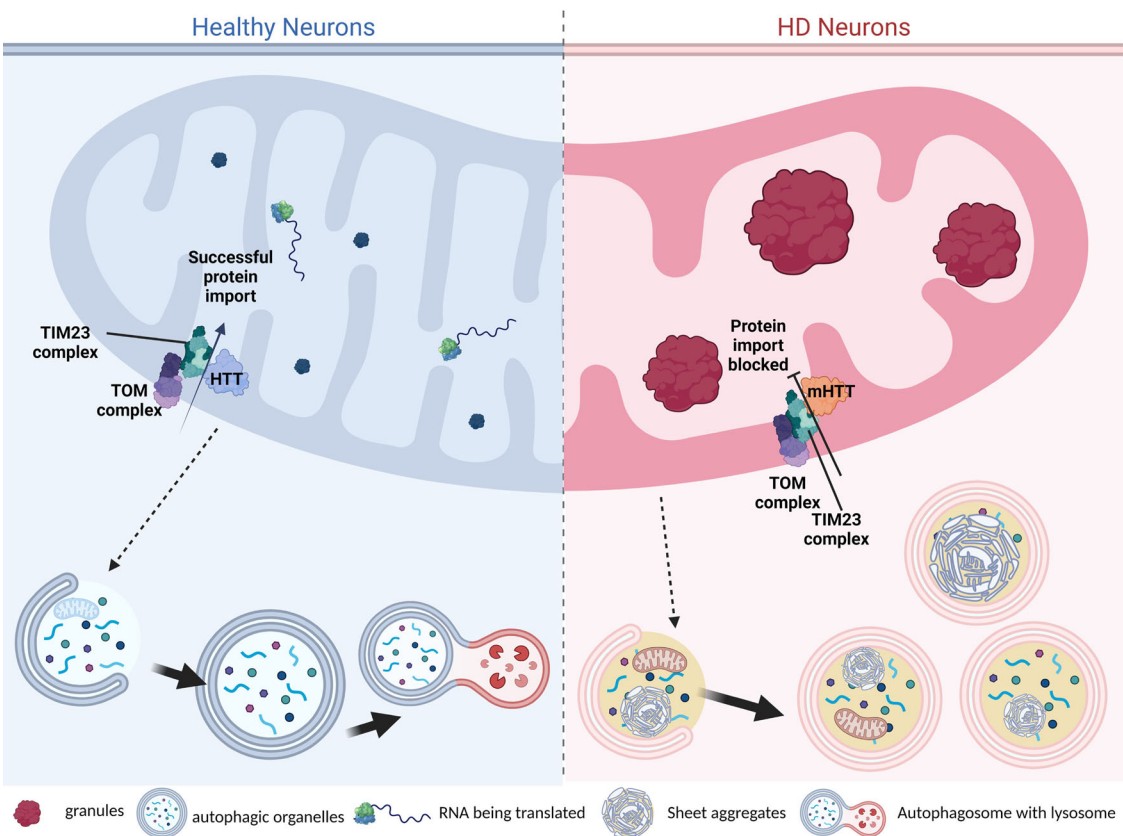

**Fig. 10 | Proposed mechanism of aberrant mitochondrial structures and sheet aggregates in autophagic organelles.** Our work here highlights the accumulation of enlarged mitochondrial granules in four human and two mouse HD neuronal models, which we hypothesize could result from disrupted protein import due to mHTT's abnormal interaction with the TIM23 complex[47]. Additionally, our mass spectrometry data suggest that RNA binding may be disrupted within HD mitochondria, which may lead to excess RNA molecules within the mitochondrial matrix. There is cross-talk between mitochondria, autophagosomes, and lysosomes, and dysfunctional mitochondria are packaged by mitophagy, in addition to other cellular waste; i.e., proteins shown within the autophagic organelles. In healthy neurons, we observe a normal autophagy cascade while in HD we observe an accumulation of sheet aggregates in autophagic organelles, which are likely not as efficiently processed for degradation as expected in healthy and WT neurons, even though we often see them associated to lysosomes. (Created in BioRender.com).

disrupt mitochondrial trafficking and impair ATP production prior to the appearance of detectable mHTT aggregates[84,85]. The N17 domain is required for the interaction of HTT with the protein import complex TIM23[46]. mHTT localizes within the intermembrane space in mitochondria and is more strongly associated with TIM23 than WT HTT, potentially blocking the import of nuclear-encoded mitochondrial proteins[47], including proteins such as GRSF1 and other RBPs. This suggests a potential mechanism (Fig. 10) underlying the aberrant structures we observe in HD mitochondria (including enlarged granules), particularly for dN17-BACHD, where normal HTT function is impaired by both the lack of the N17 domain and the expansion of the polyQ tract[34,46,52].

The sheet aggregates we observed here within autophagic organelles appear to be another early structural hallmark of HD since they were consistently present in the neurites of all HD model systems we examined. In addition to cryoET tomograms, we also collected high-magnification cryoEM images of sample areas with sheet aggregates; the lack of evidence for periodic arrangement in them by Fourier analysis suggests that they are not classical amyloid filaments with the known characteristic cross-beta sheet structures perpendicular to the filament axis[86]; however, this may not be surprising since it is possible that the sheet aggregates we observed may not contain fibrillar mHTT, and HD does not strictly fit within the diseases known as amyloidosis[87]. It is also possible that future purification of the sheet aggregates may render them more amenable to high-resolution analyses without interference from the cell membrane and surrounding milieu of the crowded cytosol. Whether the sheet aggregates contain mHTT also awaits future biochemical analyses of purified autophagic organelles and/or degenerated mitochondrial particles. Nevertheless, our data here suggest that these unusual structural features, likely in components of autophagic pathways, are a consequence of the stress and toxicity induced by the presence of full-length, endogenous mHTT or fragments derived from it.

Our observations raise the potential hypotheses that the sheet aggregates may in part represent degenerated mitochondria (Supplementary Fig. 3e, f) associated with lysosomes (Supplementary Fig. 3g); that is, the sheet aggregates may develop from remodeled and compromised mitochondrial membranes and/or granules through an unresolved mechanism in HD cells, and may go through intermediate states before the overt appearance of sheet aggregate densities within them. Indeed, we observed many double membrane-bound compartments in most of our specimens whose identities were not clearly assignable but that may correspond to mitochondria in these hypothesized intermediate states, such as the organelle in the bottom left of Supplementary Fig. 3e, and those in Supplementary Fig. 3a–d. In support of this hypothesis, mitochondria have been recently demonstrated to interact directly with lysosomes via their membranes[51]. Indeed, the traditional paradigm for largely separate functions by these two organelles has changed with emerging evidence that mitochondria and lysosomes are mutually functional and interdependent, with profound implications to the underpinnings of aging[88] and neurodegeneration[89,90].

Other challenges and limitations of our study include that we do not yet know the precise mechanism of how mitochondrial granules and sheet aggregates form or the biochemical components of the sheet aggregates, which are the focus of future studies. In spite of the challenges, our exciting observations provide an additional hint into resolving the potential origin of the sheet aggregates. Mitochondrial autophagy (mitophagy) is a clearance mechanism of defective mito-chondria via autophagy[35] and is altered in many neurodegenerative disorders such as Parkinson disease (PD) and Alzheimer disease (AD)[91]. There is now emerging evidence that mitophagy is also defective in HD. Altered mitophagy may potentially contribute to the bioenergetic deficits observed in various HD models[13,92,93]. Consistent with our findings, recent work[31] has identified non-fibrillar mHTT within single-membrane-bound organelles in cortical and striatal tissue from zQ175 HD mice, including multivesicular bodies (MVB) and amphisomes, using correlative light and electron microscopy of samples fixed by freeze substitution. The localization of non-fibrillar mHTT changed depending on the disease stage, with presymptomatic stages showing localization within MVBs/amphisomes and late-stage disease showing localization to the autolysosomes or residual bodies[31]. Our findings here in intact, cryo-preserved human patient and mouse model-derived neurons are suggestive of even earlier events and thus are complementary.

Our previous studies showed improved synaptic-associated gene expression and mitochondrial DNA integrity in HD iPSC-neurons after PIAS1 KD[36,38]. Based on these data, we evaluated the effect of reduced PIAS1 levels on the mitochondrial and autophagosome structural phenotypes in HD patient iPSC- and mouse model-derived neurons and observed rescue of these phenotypes (Figs. 8 & 9). Consistent with the rescue data, a recent report identified PIAS1 as a potential age-of-onset modifier. PIAS1 containing an S510G single nucleotide poly-morphism, which reduces SUMO modification of mHTT[94], delays HD onset, and produces milder disease severity in HD mice, consistent with the concept that reduced PIAS1 levels may ameliorate disease. Our studies thus represent a proof of concept that the synergistic combi-nation of cryoET and proteomics of iPSC- and mouse model-derived neurons or organelles within them can inform on the impact of a given therapeutic strategy on structural features and ultimately function and may be applicable to other cellular systems and disease models.

## Methods

### Ethics statement, iPSCs, and mouse models
All protocols involving the use of animals in the study, namely BACHD[32] and deltaN17 BACHD[33], were approved by the Institutional Animal Care and Use Committee at the University of California in San Diego. iPSC work was approved by UC Irvine Human Stem Cell Research Oversight Committee with patient donor consent.

### iPSC culture, differentiation, and maintenance
Information about iPSCs in Supplementary Table 3. Neuronal differ-entiation was performed once iPSC colonies reached 60–70% con-fluency as previously described[4]. Differentiation was initiated by washing iPSC colonies with phosphate-buffered saline pH 7.4 (PBS – Gibco) and switching to SLI medium (Advanced DMEM/F12 (1:1) sup-plemented with 2 mM Glutamax™ (Gibco), 2% B27 without vitamin A (Life Technologies), 10 μM SB431542, 1 μM LDN 193189 (both Stem Cell Technologies), 1.5 μM IWR1 (Tocris)) with daily medium changes. This was day 0 of differentiation; at day 4, cells were pretreated with 10 μM Y27632 dihydrochloride (Tocris) and then washed with PBS, and then passaged 1:2 with StemPro Accutase (Invitrogen) for 5 minutes at 37 °C and replated onto plates which were coated with hESC qualified matrigel (1 h at 37 °C) in SLI medium containing 10 μM Y27632 dihy-drochloride for 1 day after plating and continued daily feeding with SLI medium until day 8. At day 8, cells were passaged 1:2 as above and replated in LIA medium (Advanced DMEM/F12 (1:1) supplemented with

2 mM Glutamax™, 2% B27 without vitamin A, 0.2 μM LDN 193189, 1.5 μM IWR1, 20 ng/ml Activin A (Peprotech)) with 10 μM Y27632 dihydrochloride for 1 day after plating and daily feeding was continued through day 16. At day 16, cells were plated on either the carbon side of Quantifoil holey carbon film grids (Electron Microscopy Sciences; see below for grid preparation for cryoET), or in 6 well Nunclon coated plates with PDL and hESC matrigel for mitochondrial isolation. Cells were plated at $1 \times 10^6$ for mitochondrial isolation and at half density for cryoET of neurites in intact neurons at 500,000 per dish on the 35 mm Mat-tek glass bottomed dish containing 3 holey-carbon grids in SCM1 medium (Advanced DMEM/F12 (1:1) supplemented with 2 mM Gluta-max™, 2% B27 (Invitrogen), 10 μM DAPT, 10 μM Forskolin, 300 μM GABA, 3 μM CHIR99021, 2 μM PD 0332991 (all Tocris), to 1.8 mM CaCl₂, 200 μM ascorbic acid (Sigma-Aldrich), 10 ng/ml BDNF (Peprotech)). Cells on EM grids were topped up with an additional 1 ml of SCM1 (35 mm Mat-Tek dishes). We tried even lower densities to improve the probability of getting one cell per grid square; however, the neurons did not survive or did not differentiate well. Medium was 50% changed every 2-3 days. On day 23, medium was fully changed to SCM2 medium (Advanced DMEM/F12 (1:1): Neurobasal A (Gibco) (50:50) supple-mented with 2 mM Glutamax™), 2% B27, to 1.8 mM CaCl₂, 3 μM CHIR99021, 2 μM PD 0332991, 200 μM ascorbic acid, 10 ng/ml BDNF) and 50% medium changes were subsequently performed every 2–3 days. Cells were considered mature and ready for subsequent analyses and experiments at day 37.

### Mouse model neuronal culture and maintenance
Established protocols were followed to set up cortical neurons col-lected from mouse E18 embryos[95–97]. BACHD, WT, and dN17 mice were strain FVB/N-Tg(HTT*97Q)IXwy/J. E18 embryos of both male and female were used in the study. Briefly, cortical tissues were extracted from E18 mouse embryos and extensively rinsed in HBSS with 1% Penicillin-Streptomycin, followed by dissociation in 0.25% trypsin with 1 mg/ml DNase I. Cortical neurons were isolated and plated with plat-ing media (Neurobasal with 10% FBS, 1xB27,1xGlutaMAX) onto either glass coverslips for immunostaining or into 12 well plates for bio-chemistry at an appropriate density. Both the cover glasses and plates were pre-coated with poly-L-lysine (Invitrogen). The plating medium was replaced with a maintenance medium (Neurobasal, 1xB27, 1xGlu-taMAX) the following day. Only 2/3 of the media was replaced every other day until the conclusion of the experiments.

### Mitochondria isolation
Day 37 neurons (3 technical replicates per line, approximately 18 million neurons per replicate), were harvested using the Miltenyi MACS human mitochondria isolation kit (Miltenyi Biotec 130-094-532) with additional proteases added to the lysis buffer aprotinin (10 μg/ml), leupeptin (10 μg/ml), PMSF (1 mMl), EDTA-free protease inhibitor cocktail III (1x SIgma-Aldrich – 539134) at a concentration of 10 million/ml of lysis buffer and disruption was performed using a 27 G needle using a syr-inge to triturate 10 times up and down and then proceeding to label following the manufacturer's recommended protocol. Final mito-chondrial pellet was resuspended in 50 μl of storage buffer and flash frozen in liquid nitrogen, stored at −80 °C until mass spec analysis.

### RNA extraction & concentration
RNA extraction was performed using QIAGEN RNEasy kit following manufacturer's protocol. RNA was eluted in 50 μl of nuclease free H₂O and required further concentration for cDNA synthesis. RNA con-centration was performed using ZYMO RNA Clean and Con-centrator™–5 kit.

### cDNA synthesis
100 ng of RNA was used for cDNA synthesis using the Quantabio qScript® cDNA SuperMix kit while 25 ng of the RNA was used for RT-

 

reaction of just RNA and nuclease-free water. cDNA synthesis was performed using the manufacturer's protocol at 25 °C for 5 minutes, then 42 °C for 30 minutes, and finally 85 °C for 5 minutes in a Bio-Rad T100 thermal cycler.

### qRT-PCR

One µl of cDNA at a concentration of 3–5 ng/µl was used per reaction, with 10 µM primers for human *GRSF1* Forward TGGAGTCAGAGCA GGATGTGCA Reverse GGCGAAGATTTGACCTGCAAGC and *RPLP0* Forward TGGTCATCCAGCAGGTGTTCGA Reverse ACAGACACTGGC AACATTGCGG, mouse *Pias1* and *Eif4a2* for a housekeeping control using *Eif4a2* forward GTGGACTGGCTCACGGAGAAAA, reverse AGAA CACGGCTTGACCCTGATC *Pias1* forward CTGCACAGACTGTGACGA GATAC reverse CGCTACCTGATGCTCCAATGTG. Quantification was performed on a Quantstudio 5 using SYBR green reagent (Biorad) running delta delta CT method and calculating fold change normalizing to non-transgenic primary neurons control SMARTPool siRNA. Prism 9.0 was used to calculate statistical significance by one or two way ANOVA where appropriate.

### Western blot

Protein was harvested from frozen cell pellets (for CRISPR validation of the clone) using RIPA lysis buffer or from isolated mitochondria samples described above. Samples were subjected to SDS (sodium dodecyl sulfate) polyacrylamide gel electrophoresis and Western blotting onto nitrocellulose. Membranes were assessed using either infrared fluorescence on the Li-Cor. Antibodies were as follows: LC3ab (Cell Signaling #12741 1:500), PIAS1 (Cell Signaling #3550 1:500, Proteintech #23395-1-AP 1:1000), CTIP2 (abcam #ab18465 1:1000), ATPB (abcam #ab14730 1:10,000), normalization of loading was calculated based on REVERT total loading stain. Full blots can be viewed at the end of the **Supplementary Information** file.

### Immunofluorescence

Immunofluorescence staining on day 37 neurons was performed after fixation with 4% paraformaldehyde for 10 minutes. Cells were stored in PBS pH 7.4 at 4 °C until staining. Cells were permeabilized with 0.3% triton-X in PBS at room temperature for 10 minutes and then blocked with 2% goat serum, 3% bovine serum albumin 0.3 M glycine 0.1% triton-X PBS for 1 h at room temperature. Primary antibody incubation was done overnight at 4 °C in block using the following antibodies DARPP32 (abcam ab40801 1:200) and CTIP2 (abcam ab18465 1:500) and rabbit IgG and rat IgG2a istotype controls (1:500). Cells were washed three times with PBS before incubation with secondary antibodies Alexa Fluor goat anti rabbit IgG 555 and rat IgG 488 for 1 h in the dark at room temperature. After incubation, cells were washed three times with PBS and incubated with Hoechst 33342 (1:5000) for 10 minutes, cells were washed a further two times and then mounted with Fluoromount G. Images were taken using the Keyence BZ-X810 Widefield Microscope using a 20x objective.

### Accell siRNA knockdown in neurons

Knockdown of *GRSF1* was performed in iPSC-neurons using Accell SMARTPool siRNA against *GRSF1* (Horizon Discovery cat#E-011677-00-0010) and non-targeting SMARTPool control (Horizon Discovery cat#D-001910-10-05) for comparison. Treatment was performed on day 28 at a concentration of 1 µM in 2 ml of medium per 35 mm MatTek dish. Half media changes were performed at day 30 and then every 2-3 days thereafter.

Knockdown of *Pias1* was performed for mouse model neurons using Accell SMARTPool siRNA against *Pias1* (Horizon Discovery cat#E-059344-00-0005) and a non-targeting SMARTPool control (Horizon Discovery cat#D-001910-10-05) for comparison. Treatment was performed at 3 days in vitro (DIV3) at a concentration of 1 µM in 1 ml of medium per 35 mm MatTek dish, the medium was topped up to 2 ml

after 24 h and then 2/3 of the media was replaced every other day until the conclusion of the experiments.

### CRISPR-Cas9 heterozygous knockout of PIAS1 in iPSC

For the 66Q line, clones were selected from stem cell edited pools in a method that was previously described[36]. Briefly, cells at 70% confluence, CS02iHD66n4 iPSCs were pre-treated with 10 µM Y27632 dihydrochloride for 1 h prior to transfection with the CRISPR/CAS9 RNP complex and a pEF1α-puromycin plasmid. CRISPR-guide RNA CAS9 ribonucleic acid protein complex was made following manufacturers instructions at 1 µM of Alt-R CRISPR-CAS9 crRNA sequence GCGTCCGTGCTTGTTTCTCC for targeting the PIAS1 locus and the Alt-R tracR RNA. A single cell suspension in Nucleofector solution II for hESC and 1 µg of pmaxGFP plasmid (Lonza), 1 µg of pEF1α-puro and 5 µl of the RNP complex were nucleofected using program B-016. Cells were subsequently cultured in mTESR supplemented with CloneR (Stem Cell Technologies) 1 µM azidothymidine (AZT - Tocris) and 10 µM Y27632 dihydrochloride on hESC-qualified Matrigel. One day after transfection, AZT was removed, and cells were treated with 200 ng/ml puromycin for 48 h with a daily medium change. Clones were picked 24 h after puromycin was removed by treating cells for 30 seconds with Versene at 37 °C and then colonies were picked using a P200 tip and re-plated in mTESR1 supplemented with CloneR. Cells were maintained in CloneR for 2 more days and then maintained in mTESR1 to collect genomic DNA. To validate genomic editing, genomic DNA was extracted from the iPSCs using the QIAGEN genomic DNA extraction kit (Qiagen). PCR across the region of interest containing potential in-dels was performed using the KAPA Hi-Fi PCR kit (KAPA Biosystems) following the manufacturer's protocol using CTCCT GGAGACTCAGTAAGTGC (forward) TAATGGCGATGATGCAGGGT (reverse). PCR products were sequenced at Eurofins using the following primers TGCCATCTACATAGCACTCGAC and GACAAGTCTGCAGG CGTCAT.

For the 53Q PIAS1 knockdown line was generated by the delivery of a CRISPR-Cas9 ribonucleoprotein (RNP) complex containing gRNA targeting PIAS1 (TCTTCAGAGGTTACGAGCAA) and high-fidelity (HiFi) Cas9 protein (IDT) at a molar ratio of 1.2:1.0 gRNA:Cas9. The RNP complex was introduced into $2 \times 10^5$ iPSCs by nucleofection, using Nucleofector Solution Kit 1 (Lonza) with program B-016. Transfected cells were allowed to recover overnight, then dissociated for single-cell plating in 96-well plates. Clonal colonies were visually identified for expansion and PCR-based screening using primers GATCATCTCGG CAGCTTTTTGG (Forward) and GCCAAGATGCCCTTCCCATTTC (Reverse), spanning the targeted region. PCR products positive for editing, as determined by Sanger sequencing, were subject to TOPO TA cloning (Life Technologies) to allow for the sequencing of each allele independently. Clones that showed heterozygous editing of PIAS1 were chosen for further analysis, including confirmation of genomic integrity by aCGH analysis (Cell Line Genetics) and reduction of PIAS1 protein by Western.

### Grid preparation for cryoET of HD patient iPSC-neurons and mouse model neurons

For iPSCs, Quantifoil® R 2/2 Micromachined Holey Carbon grid: 200 mesh gold (SPI supplies Cat#:4420G-XA) grids were prepared for cell plating by sterilizing using forceps to carefully submerge them in 100% ethanol (Fisher Scientific) at an angle perpendicular to the liquid surface and then passed quickly through a yellow flame. Grids were then placed into 1 ml of poly-D-lysine (100 µg/ml in borate buffer, pH 8.4) in a 35 mm Mat-Tek glass coverslip bottom dish (VWR P35G-0-14-C) at an angle perpendicular to the liquid's surface, and coated on the bottom of the dish for at least 1 h at room temperature. When cells were ready for plating, the grids were washed two times with sterile-filtered Milli-Q $H_2O$ and a final wash with PBS before cell plating.

For mouse neurons, EM grids were briefly dipped into 70% ETOH to sterilize, followed by coating with 0.1 mg/ml PDL[98]. The grids were rinsed with sterile dH₂O and loaded with isolated neurons. The maintenance of these grids was exactly as described above. DIV14 neurons were used for cryoEM/cryoET experiments.

For both iPSC-neuron and mouse model neurons, cells grown on grids were vitrified using the temperature- and humidity-controlled Leica GP2. Grids were retrieved from culture dishes using the forceps for the Leica device, and 3 μl of 15 nm BSA gold tracer (Fisher Scientific; Catalog No.50-248-07) was pipetted onto the carbon- and cell-side of the grid. Grids were loaded and blotted for 5 seconds in 95% humidity at 37 °C and immediately plunged into liquid propane. The vitrified grids were transferred into grid storage boxes in clean liquid nitrogen and then stored in clean liquid nitrogen prior to cryoET imaging.

Cells on the remainder of the dishes that were used for cryoET, were scraped using a cell scraper in cold PBS and pelleted at 350 xg for 5 minutes, PBS was aspirated and the pellets were flash-frozen in liquid nitrogen and stored at −80 °C for later analysis of knockdown.

### CryoEM/ET data collection
For each specimen, namely vitrified HD patient iPSC-neurons and mouse primary neurons, we collected low magnification (6500X) cryogenic TEM images and assembled them into montages to screen for regions of interest (ROI) before cryoET tilt series collection. Images were acquired using a G3 Titan Krios™ or Talos Arctica cryo-electron microscope (ThermoFisher Scientific) operated at 300 or 200 kV, energy filter at 20 eV, in low-dose mode using SerialEM software[99]. At each tilt angle, we recorded "movies" with 5−6 frames per movie using a Gatan K2™ or K3™ direct electron detection camera with a Bio-Quantum™ Imaging Filter (Gatan, Inc). The tilt series were collected at a magnification corresponding to 3.47 Å/pixel using a dose-symmetric tilting scheme[100] from 0°, target defocus of −5 μm and a cumulative dose of ~120 e/Å².

### Tomographic reconstruction
Upon data collection, all tilt series were automatically transferred to our computing and storage data clusters and images were motion-corrected using MotionCor2[101] and reconstructed into full tomograms automatically using EMAN2[102]. This on-the-fly reconstruction facilitated the screening of tomograms.

After screening the automated cryoET reconstructions, we used IMOD[103] software for standard weighted-back projection tomographic reconstruction of most tomograms with interesting and relevant features, using coarse cross-correlation-based alignment, gold fiducial-based alignment, or patch tracking alignment. For each tilt series, unsuitable images with large drift, excessive ice contamination, etc. were manually removed prior to tilt series alignment. For cryoET tilt series containing prominent sheet aggregates and subjected to Fourier analysis and subtomogram averaging attempts to determine whether they contained repeating features, we corrected for the contrast transfer function (CTF) using IMOD's recent 3D-CTF correction algorithm and reconstructed them using a SIRT-like filter (16 iterations) (Fig. 4 and Supplementary Fig. 5).

### Tomographic annotation and segmentation
The tomograms containing mitochondrial granules were post-processed binning by 4x and applying various filters, such as a low-pass Gaussian filter at a frequency = 0.01, a Gaussian high-pass filter to dampen the first 1−5 Fourier pixels, normalization, and thresholding at 3 standard deviations away from the mean. For both types of tomograms (containing mitochondrial granules or sheet aggregates), we carried out tomographic annotation of different features in binned-by-4 tomograms using the EMAN2 semi-automated 2D neural network-based annotation[43], and performed manual clean-up of false positives

in UCSF Chimera[104]. The cleaned-up annotations were thresholded, low-pass filtered, and turned into binary masks, which were multiplied by contrast-reversed versions of the tomograms to segment out each corresponding feature.

### Visualization of tomograms containing sheet aggregates
Initially, the sheet aggregates appeared filamentous in 2D z-slices through our cryoET tomograms. When attempts at subtomogram averaging failed to yield averages with filamentous morphology, and instead resulted in featureless sheets, we more carefully examined the tomograms slice-by-slice to understand the causes for this apparent failure. This revealed that the linear features in a section persisted through multiple sections above or below. While this might be expected due to lower resolvability in z because of the missing wedge, we also noticed that the linear features shifted their location from section to section (as if drifting sideways), and in different directions. This anomaly suggested that the linear features seen in 2D slices corresponded to sheets in 3D, oriented at various angles with respect to the x-y plane, and explained the preliminary exploratory subtomogram averages that had also revealed a sheet morphology. It is of note that the visualization of these sheets was greatly enhanced by the use of UCSF ChimeraX's virtual reality (VR) 3D stereo graphics[105], which was used to help in the production of Supplementary Movie 2.

The sheets have high contrast in the tomograms, but are exceedingly thin (≤20 Å, thinner than lipid bilayers in the same tomograms), as measured from 3D-CTF corrected tomograms without downsampling or low-pass filtration. They are only visible if they are oriented such that at least some of the images in the tilt series view them parallel to their surfaces; otherwise, the missing wedge renders them invisible.

### Quantification and statistical analysis of mitochondria granules size
Even though recent protocols can dramatically increase the speed of manual annotation[106], manual methods suffer from human bias and large inconsistencies across annotators and features[107]. Furthermore, popular semi-automated methods[43] are geared mainly towards visualization and have not been demonstrated or streamlined for feature quantification. Therefore, we developed a two-stage deep learning system for voxel level annotation of mitochondria and granules in tomograms for purposes of automated quantification of mitochondria and mitochondrial granule numbers and volumes. In the first stage, our system is trained on a handful of annotated slices on a subset of tomograms and learns to segment mitochondria and granules. In the second stage, we use our model to make predictions on all the tomograms and use high confidence predictions as pseudo-annotations to augment our training set. We then train a new model on this augmented training set and use its predictions to quantify the number and sizes of mitochondria and granules in the tomogram.

In the first stage, we train a 3D-UNet[108] model to perform segmentation of the 3D volumes containing objects of interest. We train two separate models, one for segmenting mitochondria and the other for segmenting granules. These models are trained in a semi-supervised fashion on sparsely annotated 3D volumes - only 2% of the 2D slices are manually annotated with each pixel being labeled as being part of the background or being part of the mitochondria/granule.

Since annotating each slice is a time-consuming process, we utilize pseudo-labelling to generate more annotations. In the second stage, we run every tomogram through our model and add the high-confidence predictions to the training set. Next, we retrain our model on the expanded training set which consists of both human- and machine (pseudo)-labeled slices. The new model is then used to make the final voxel level predictions on all the tomograms.

We refine the 3D segmentations in the post-processing stage. We run a connected components analysis to count the number of segmented objects. We filter out background noise in the predictions by discarding objects which don't fall within the expected range of volumes. We retain granule detections which are located within a detected mitochondrion. After post-processing which included further binning, we scale each voxel accordingly to get the volume in nm³.

Compilation of the data into graphs was performed in Prism 9.3 (https://www.graphpad.com/scientific-software/prism/) with statistics performed based on the number of groups to compare and data normality. iPSC-neurons used Dunn's multiple comparisons to compare control to the various HD lines, for the mouse primary neurons, Dunn's multiple comparisons were used to compare WT vs all other groups and then BACHD vs dN17 and Control siRNA vs *Pias1* siRNA treatment.

### Sample preparation for proteomic analysis

Isolated mitochondria were solubilized in a final concentration of 1% SDS and mitochondrial proteome was extracted using methanol-chloroform precipitation. 400 μl methanol, 100 μl chloroform and 350 μl water were added sequentially to each 100 μl sample, followed by centrifugation at 14,000 x *g* for 5 min at room temperature. The top phase was removed, and the protein interphase was precipitated by addition of 400 μl methanol, followed by centrifugation at 14,000 g for 5 min at room temperature. Pellet was air dried and resuspended in 8 M urea, 25 mM ammonium bicarbonate (pH 7.5). Protein concentration was determined by BCA (Thermo Fisher) and 2–4 μg total protein were subjected to reduction and alkylation by incubation with 5 mM DTT for 1 h at room temperature followed by 5 mM iodoacetamide for 45 min at room temperature, in the dark. The samples were then incubated with 1:50 enzyme-to-protein ratio of sequencing-grade trypsin (Promega) overnight at 37 °C. Peptides were acidified with trifluoroacetic acid to a final concentration of 1%, desalted with μC18 Ziptips (Millipore Sigma), dried, and resuspended in 10 μL 0.1% formic acid in water.

### LC-MS/MS acquisition

LC-MS/MS analyses were conducted using a QExactive Plus Orbitrap (QE) mass spectrometer (Thermo Fisher) coupled online to a nanoAcquity UPLC system (Waters Corporation) through an EASY-Spray nanoESI ion source (Thermo Fisher). Peptides were loaded onto an EASY-Spray column (75 μm x 15 cm column packed with 3 μm, 100 Å PepMap C18 resin) at 2% B (0.1% formic acid in acetonitrile) for 20 min at a flow rate of 600 nl/min. Peptides were separated at 400 nL/min using a gradient from 2% to 25% B over 48 min (QE) followed by a second gradient from 25% to 37% B over 8 minutes and then a column wash at 75% B and reequilibration at 2% B. Precursor scans were acquired in the Orbitrap analyzer (350–1500 m/z, resolution: 70,000@200 m/z, AGC target: $3 \times 10^6$). The top 10 most intense, doubly charged or higher ions were isolated (4 m/z isolation window), subjected to high-energy collisional dissociation (25 NCE), and the product ions measured in the Orbitrap analyzer (17,500@200 m/z, AGC target: 5e4).

### Mass spectrometry data processing

Raw MS data were processed using MaxQuant version 1.6.7.0 (Cox and Mann, 2008). MS/MS spectra searches were performed using the Andromeda search engine[109] against the forward and reverse human and mouse Uniprot databases (downloaded August 28, 2017 and November 25, 2020, respectively). Cysteine carbamidomethylation was chosen as fixed modification and methionine oxidation and N-terminal acetylation as variable modifications. Parent peptides and fragment ions were searched with maximal mass deviation of 6 and 20 ppm, respectively. Mass recalibration was performed with a window of 20 ppm. The maximum allowed false discovery rate (FDR) was <0.01 at both the peptide and protein levels, based on a standard target-decoy database approach. The "calculate peak properties" and "match between runs" options were enabled.

All statistical tests were performed with Perseus version 1.6.7.0 using either ProteinGroups or Peptides output tables from MaxQuant. Potential contaminants, proteins identified in the reverse dataset, and proteins only identified by site were filtered out. Intensity-based absolute quantification (iBAQ) was used to estimate absolute protein abundance. Two-sided Student's *t*-test with a permutation-based FDR of 0.01 and S0 of 0.1 with 250 randomizations was used to determine statistically significant differences between grouped replicates. Categorical annotation was based on Gene Ontology Biological Process (GOBP), Molecular Function (GOMF) and Cellular Component (GOCC), as well as protein complex assembly by CORUM.

Additional analysis was performed on only the unique peptides of the differentially enriched or depleted proteins that were significant by *t*-test, using Panther pathways and Panther Overrepresentation algorithms for GO Molecular Function, GO Biological Processes and GO Cellular Component at http://www.pantherdb.org/. Ingenuity Pathway Analysis was performed using the significantly differential proteins to assess pathways, networks and upstream regulators. For comparison of DEPs to PIAS1 knockdown DEGs, overlap statistics for overrepresentation was performed at http://nemates.org/MA/progs/overlap_stats.html using the total genome number of genes at 20,500.

### Reporting summary

Further information on research design is available in the Nature Portfolio Reporting Summary linked to this article.

### Data availability

Representative tomograms for each phenotype and condition investigated in this study have been deposited in the EMDB (https://www.emdataresource.org/deposit.html). The accession codes are as follows: EMD-28668 (iPSC Q18 neuron with enlarged mitochondrial granules), EMD-28944 (iPSC Q20 neuron with enlarged mitochondrial granules), EMD-28946 (iPSC Q53 neuron with enlarged mitochondrial granules), EMD-29074 (iPSC Q66 neuron with enlarged mitochondrial granules), EMD-29075 (iPSC Q77 neuron with enlarged mitochondrial granules.), EMD-29076 (iPSC Q109 neuron with enlarged mitochondrial granules), EMD-29080 (iPSC Q66 neuron with PIAS1-hetKO treatment showing normal mitochondrial granules), EMD-29081 (iPSC Q66 with GRSF1-KD treatment showing normal mitochondrial granules), EMD-29210 (mitochondria purified from iPSC Q109 neuron with enlarged mitochondrial granules), EMD-29083 (WT mouse neuron showing normal mitochondria), EMD-29207 (BACHD mouse model neuron with enlarged mitochondrial granules), EMD-29084 (BACHD-dN17 mouse model neuron with enlarged mitochondrial granules), EMD-29079 (iPSC Q66 neuron with sheet aggregates in autophagic organelles), EMD-29211 (iPSC Q66 neuron with PIAS1-hetKO treatment rescues sheet aggregates in autophagic organelles) and EMD-29208 (BACHD mouse model neuron with sheet aggregates in autophagic organelles). The mass spectrometry proteomics data generated in this study have been deposited to the ProteomeXchange Consortium via the PRIDE partner repository with the dataset identifier PXD037526. Source data are provided with this paper.

### Code availability

The software and AI models we developed to segment the mitochondria and granules are accessible at GitHub (https://github.com/sanketx/mitochondria_segmentation).

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

## Acknowledgements

We thank Dr. William X. Yang's group at University of California in Los Angeles for providing the BACHD and deltaN17-BACHD mouse models, Weijiang Zhou for feedback on analysis of high-resolution images of sheet aggregates to look for possible repeating features, and Ronald Courville for assistance with generating manual labels to train the artificial intelligence algorithm to find and quantify mitochondrial granules. We thank Dr. Lisa Salazar and Amber Keith and the UCI SCRC CRISPR Core for generation and validation of the 53Q PIAS1hetKO cell line. We thank the support of NIH grants (P01NS092525 to LMT, WC, JF, WM; S10OD021600 to WC; R35NS116872 to LMT), postdoctoral fellowships from the Hereditary Disease Foundation to GHW and CSG, HDSA Human Biology Project to CSG, and Chan Zuckerberg Initiative Neurodegeneration Challenge Pairs Pilot Project to WC and SY (2020-221724, 5022).

## Author contributions

G.-H.W., C.S-G., J.G.G.-M., P.M., L.M.J., M.F.S, C.W., W.M., J.F., L.M.T., and W.C. were involved in conception and design of the experiments related to differentiation and cryo electron tomography. G-H.W., J.G.G.-M., M.F.S., and W.C. analyzed cryo-ET data. C.S-G. optimized iPSC cell growth on EM grids, and performed all iPSC differentiations and cell cultures. Y.G. performed all mouse model neuronal cultures. G.-H.W., L.M.J., and P.M. optimized cryoET grid preparation and screening and collected all cryoET data. J.G.G.-M., and G-H.W. performed tilt series alignment and tomographic reconstruction. J.G.G.-M., G.-H.W., and C.D. performed tomographic annotation. J.G.G.-M., G-H.W., and M.F.S. participated in cryoET data visualization. C.S-G, R.A., J.F., and L.M.T. were involved in the conception and design of the experiments for the mitochondria mass spectrometry. C.S-G, R.A., N.R.G., and L.M.T. were involved in the acquisition, the validation, analysis, and interpretation of the mitochondrial mass spectrometry data. C.S.-G, Y.G., R.M., and K.Q.W. were involved in the differentiation and cell culture of the neurons and validation of PIAS1 knockdown. S.G., J.H., and S.Y were involved in creating an artificial intelligence-based algorithm for the automated segmentation of features in cryoET tomograms and quantitative analyses of mitochondrial granules. G.-H.W., J.G.G.-M., C.H., and C.D. were involved in extensive manual reference labeling to train artificial intelligence algorithms for the segmentation and quantification of features in cryoET tomograms. J.G.G.-M., C.S.-G., and G.-H.W. prepared the final figures and movies. G.-H.W, C.S-G, J.G.G-M., R.A., C.W., S.G., S.Y., L.M.T., and W.C, wrote the manuscript. G.-H.W, C.S-G, J.G.G-M., M.F.S., L.M.T., and W.C., substantively revised the manuscript with input from all authors.

## Competing interests

The authors declare no competing interests.

## Additional information

[1]Department of Bioengineering, James H. Clark Center, Stanford University, Stanford, CA 94305, USA. [2]Department of Psychiatry & Human Behavior University of California Irvine, Irvine, CA 92697, USA. [3]Department of Neurosciences, University of California San Diego, La Jolla, CA 92037-0662, USA. [4]Department of Computer Science, Stanford University, Stanford, CA 94305, USA. [5]Department of Biology, Stanford University, Stanford, CA 94305, USA. [6]Division of CryoEM and Bioimaging, SSRL, SLAC National Accelerator Laboratory, Stanford University, Menlo Park, CA 94025, USA. [7]Department of Memory Impairment and Neurological Disorders, University of California Irvine, Irvine, CA 92697, USA. [8]Department of Neurobiology and Behavior, University of California Irvine, Irvine, CA 96267, USA. [9]Department of Biomedical Data Science, Stanford University, Stanford, CA 94305, USA. [10]Department of Genetics, Stanford University, Stanford, CA 94305, USA. [11]Sue & Bill Gross Stem Cell Research Center, University of California Irvine, Irvine, CA 96267, USA. [12]Department of Biological Chemistry, University of California Irvine, Irvine, CA 92617, USA. [13]Department of Microbiology and Immunology, Stanford University, Stanford, CA 94305, USA. [14]These authors contributed equally: Gong-Her Wu, Charlene Smith-Geater. [15]These authors jointly supervised this work: Leslie M. Thompson, Wah Chiu. ✉e-mail: lmthomps@uci.edu; wahc@stanford.edu

