## [Peer Review File · Nature Communications]

CryoET Reveals Organelle Phenotypes in Huntington Disease Patient iPSC-Derived and Mouse Primary NeuronsREVIEWER COMMENTS

Reviewer #1 (Remarks to the Author):

The authors have applied cryogenic electron tomography in an attempt to identify early pathological biomarkers in neurites from iPSC-derived neurons from Huntington's disease (HD) patients with varying CAG repeat expansions. In the HD lines, they discovered two types of aberrant structures: mitochondria with enlarged granules and distorted cristae, and thin sheet aggregates in double membrane-bound organelles. They applied the same analysis to primary cortical neurons from BACHD, deltaN17-BACHD, and wild-type mice, which confirmed that these structures were a consequence of the HD mutation. A comparison of the mitochondrial proteomic analysis from HD and control iPSC-derived neurons with previous datasets that they had generated led them to propose that a reduction in PIAS1 might improve the mitochondrial dysmorphologies. Consistent with their hypothesis, knockdown of PIAS1 in the iPSC-derived neurons and BACHD primary neurons partially rescued the mitochondrial phenotypes, data confirmed for the iPSC-neurons by an AI-based quantification of the mitochondrial granules.

This paper is a technical tour de force that has succeeded in identified novel pathological phenotypes in HD iPSC-neurons and will be of interest to the wider neurodegenerative community. The authors extensively discuss the likely composition of the granules and sheet-aggregates, further investigation of which is beyond the scope of this paper. There are some minor comments to address.

There is little information as to the source of the iPSCs. Why did the applicants choose to use only one control line? Was it because the lines are isogenic? Having an n = 1 for one of the groups is not ideal, and without the mouse data, this would be a limitation.

The number of replicates is not mentioned. Supplementary Table 1 lists the number of EM grids. Would this correspond to the number of technical replicates?

In Fig 1, the mitochondrial granules in Q66 do not look enlarged, in contrast to those in Q109. Do the authors believe their surprising quantitative data showing in Fig. 9 suggesting that there was no difference between granule volume in the Q18 and Q109 lines? In line 472 they confirm that they manually visualized hundreds of neurite tomograms – did their impression match with the AI data?

Fig 2 shows the structure of the granules in a Q77 neuron. What was the structure of the granules in the Q18 neurons? Were they similar but smaller?

At various stages throughout the manuscript, the authors speculate about the nature of the sheet-like aggregates. They discuss the possibility that they are composed of mHTT, or that they may be degenerating mitochondria associated with lysosomes. Would it be possible to perform immunoEM with antibodies to HTT to determine whether they contain HTT?

Line 287 – were the proteomic analyses performed on technical replicates for each of the two lines? At the end of the legend to Figure 7, it is stated that $n = 3$. What does this refer to?

Minor edits required:

Line 63: the referencing for the presence of inclusions in mouse models and HD post-mortem brain is mixed up.

Line 333 – Should be Supplementary Fig 4c-f.

Line 363 – Fig 7d in the wrong place in the sentence

Lines 431-434 – Fig. 9b and 9d are not mentioned in the text.

Supplementary Fig. 1: abbreviations should be defined

Supplementary Fig 2: what does the blue box represent?

Supplementary Fig 3: f is missing from the legend

Reviewer #2 (Remarks to the Author):

The presented manuscript is technically sound. The statistical analyses of granules and sheet aggregates in various cell types are properly conducted. The main findings are that both granules and sheet aggregates are altered in the HD cells, which maybe partially rescued by Pias1 KD or KO. Proteomic analysis points to a number of processes/pathways, yet the mechanism is still unclear. The identity of

the membrane-bound compartments containing sheet aggregates (lysosomes, autophagosomes etc) could have been identified or defined by using a CLEM approach. Similarly, the hypothesis of mitochondria-lysosome interaction for the inclusion of sheet aggregates in the membrane-bound compartment could be tested using the CLEM approach.

Specific comments:

1. There are sheet-like aggregates (Poly-Gly-Ala) and other aggregates observed in neurons in previous studies, which are relevant to this study and should be discussed and cited.

Q. Guo et al. In situ structure of neuronal C9orf72 poly-GA aggregates reveals proteasome recruitment Cell, 172 (2018), pp. 696-705.e12)

Riemenschneider, H. et al.. EMBO reports. e53890 2022

Gel-live inclusion of C-terminal fragments of TDP-43 sequester stalled proteasome in neurons

2. Due to sample thickness, the authors can only survey structural features in the thin neurites. Would the observed changes also occur throughout the cell body?

3. Line153-155: Could the authors quantify the “abnormal” cristae?

4. Fig5, to quantify the abundance of the sheet-like aggregates in autophagic organelles and compare them between different cell types, could the authors provide the percentage of autophagic organelles containing sheet-like aggregates, and also calculate the volume ratio between aggregates and autophagic organelle from different cell types?

5. Fig S2, what is the blue box?

Reviewer #3 (Remarks to the Author):

This manuscript discovered HD-related mitochondrial phenotypes in neurons using a combination of CryoET and proteomics approach. Mitochondrial and autophagosome morphology from multiple hiPSC-neuron lines and primary mouse neurons expressing human mHTT were examined in this study. Enlarged granules inside mitochondria are striking phenotype which was rescued or partially alleviated by PIAS1 KD. AI-based imaging processing also enabled the quantification of mitochondrial phenotypes which can be a useful tool for the field. Detailed comments are listed below:

1. In Page 7, the author mentioned that the mitochondrial phenotypes became more pronounced with higher polyQ length. Please provide the CAG repeat lengths for these five difference Ipsc-neuron lines and provide a correlation analysis to prove the claim that higher polyQ length neurons have severer phenotypes.

Also more information needs to be provided for these five iPSC lines. Are they from different patients? And basic patient information like age, sex, disease status et al.

2. Both hiPSC-neurons and mouse primary neurons showed enlarged granules and abnormal cristae in neurites. Are this abnormal morphology only present in neurites, what about cell body? Or is HD relevant changes mainly appearing in neurites supported by the literature?

3. Can this high resolution cryoET differentiate dendrites vs. axons?

4. Figure 6a shows isolated mitochondria using anti-TOM22 microbeads. Are these two images only from the Q109 HD neurons? Can the author also show the one from control neurons? The morphology differences in patient vs. ctrl neurons may influence their efficiency of being captured by the microbeads and therefore influencing their proteome composition/abundances.

5. It would be more convincing if the author can conduct proteomics on the rescued neurons compared to the HD neurons. Many of these changed proteins are high abundant mitochondrial proteins which should be detectable in even whole neuron proteomics without the need to isolate mitochondria.

6. In Page 24. The author mentioned previous studies that identified mitochondrial dysfunction in HD neurons. Do authors see these relevant protein changes in the current proteomics dataset? For example, changes in OXPHOS complexes in the isolated mitochondria?

Minor comments:

- How many replicates (dish of neurons) were conducted for each group in the proteomics experiment? In the method section for mitochondrial isolation. Also, can the author indicate how many neurons in what scale of culture dish are sufficient to provide enough isolated mitochondria for proteomics?
- In page 14, the author mentioned 177 differentially enriched peptides with a potential 236 identities. Do you mean 177 peptides belong to 236 protein groups were changing? Is the quantification performed in the protein level or peptide level?
- Since the paper mainly focused on mitochondria and autophagosome morphology, maybe consider revising the title to be more specific to these two organelles or just mitochondria.

Reviewer #4 (Remarks to the Author):

Thank you very much for asking me to review this manuscript from Wu et al. This is an interesting observational report which identifies intramitochondrial aggregates in the neurites of iPSC-neurons derived from patients and from mouse HD models. The numbers of granules/aggregates in the mito populations generally increased as a function of polyQ repeat, although this correlation was not complete. Surprisingly, these unusual aggregates could be partially ameliorated by depletion of a sumoylating protein PIAS1. Whilst these observations are important and should be published I feel the paper is rather phenomenological and is a bit frustrating. The authors report these interesting series of aggregates/granules and make some suggestions as to what they may be (RNA granules ?, CaP precipitates ?) but do not follow this up. I agree this is not trivial, but it would strengthen the paper substantially if the authors knew what they were. I am also far from convinced by the authors overall conclusion that these aggregates may be due to aberrant mito protein import, as the authors suggest an association between mHTT and a component of the TIM complex which could lead to aberrant import. The authors seem to have data that contradicts this suggestion, as differentially expressed proteins identified by MS show very little evidence of a tendency towards loss of mito matrix proteins. The authors suggest that the increase in granules may actually be indications of increased RNA granules, although a key member of the RNA granule, GRSF1 is actually downregulated. As I mentioned, identifying what constitute these granules is really important. Determining whether they are due to increased RNA aggregation should be very simple, as the authors could just perform a mitochondrial RNA:FISH expt and visualise by standard confocal microscopy. There are only 11 mt-mRNA species and 2 mt-rRNA species so it is quite simple to assess. Overall, the authors are no doubt and understandably

very proud of their methodology, but I feel the paper is driven more by this methodology rather than the authors being driven by trying to identify what these aggregates are and how they are formed.

October 24, 2022

To: Dr. Katarzyna Marcinkiewicz, Senior Editor, Nature Communications

From: Dr. Wah Chiu, Professor, Stanford University

Re: Responses to the Reviewers' Comments for Manuscript NCOMMS-22-11340

Dear Dr. Marcinkiewicz,

We are grateful for your invitation to revise our manuscript. Please see our point-by-point detailed response to the Reviewer's useful feedback below. We have extensively modified our manuscript accordingly and throughout, hence are submitting a clean version. We have indicated where edits were made below.

REVIEWER COMMENTS

Reviewer #1 (Remarks to the Author):

1. The authors have applied cryogenic electron tomography in an attempt to identify early pathological biomarkers in neurites from iPSC-derived neurons from Huntington's disease (HD) patients with varying CAG repeat expansions. In the HD lines, they discovered two types of aberrant structures: mitochondria with enlarged granules and distorted cristae, and thin sheet aggregates in double membrane-bound organelles. They applied the same analysis to primary cortical neurons from BACHD, deltaN17-BACHD, and wild-type mice, which confirmed that these structures were a consequence of the HD mutation. A comparison of the mitochondrial proteomic analysis from HD and control iPSC-derived neurons with previous datasets that they had generated led them to propose that a reduction in PIAS1 might improve the mitochondrial dysmorphologies. Consistent with their hypothesis, knockdown of PIAS1 in the iPSC-derived neurons and BACHD primary neurons partially rescued the mitochondrial phenotypes, data confirmed for the iPSC-neurons by an AI-based quantification of the mitochondrial granules.

This paper is a technical tour de force that has succeeded in identified novel pathological phenotypes in HD iPSC-neurons and will be of interest to the wider neurodegenerative community. The authors extensively discuss the likely composition of the granules and sheet-aggregates, further investigation of which is beyond the scope of this paper. There are some minor comments to address.

We thank the Reviewer for their positive feedback.

There is little information as to the source of the iPSCs. Why did the applicants choose to use only one control line? Was it because the lines are isogenic? Having an $n = 1$ for one of the groups is not ideal, and without the mouse data, this would be a limitation.

At the Reviewer's suggestion, we have added **Supplementary Table 4**, detailing further information on the source of iPSCs and included further references that also document this in detail. We have added a second control subject line with Q=20 and have analyzed iPSC-derived neurons (hereafter, "iPSC-neurons") from this control. After extensively screening the neurites of neurons grown on 3 different cryoEM grids, we failed to find mitochondria with enlarged granules or double-membrane-bound sheet aggregates in the Q20 line. Our AI-based analyses confirmed that the mitochondrial granules were indeed "normal", as were those from the other control cell line, Q18. We have updated Figs. 1 and 9 and their corresponding legends to include this additional control (Q20), which reinforces our conclusions.

2. The number of replicates is not mentioned. Supplementary Table 1 lists the number of EM grids. Would this correspond to the number of technical replicates?

Yes. To clarify the entries in Supplementary Table 1, we have reorganized them to explicitly state for each cell type the number of EM grids, representing technical replicates, the number of tomograms reconstructed, and how many of them contain mitochondria or sheet aggregates in phagocytic vesicles.

3. In Fig 1, the mitochondrial granules in Q66 do not look enlarged, in contrast to those in Q109. Do the authors believe their surprising quantitative data showing in Fig. 9 suggesting that there was no difference between granule volume in the Q18 and Q109 lines? In line 472 they confirm that they manually visualized hundreds of neurite tomograms – did their impression match with the AI data?

As shown in Fig. 9, each cell line exhibits a spectrum of mitochondrial granule sizes, likely because the mitochondria were randomly chosen and may be in different physiological states. We agree that it is better to show a more representative tomogram for Q66 with enlarged mitochondrial granules to reflect the quantitative conclusions derived from population statistics about granule sizes. See revised Fig. 1.

Regarding there being no statistical difference between Q18 (and now Q20) and Q109 in terms of granule size, this was a surprising result for us as well. Our qualitative observations indeed match the results from the AI-based statistical quantification. While we are not certain as to the biological causes for this phenomenon, one hypothesis mentioned in the Discussion is as follows: Mitochondria in Q109 iPSC-neurons may be in a later degeneration stage where the enlarged granules have dissolved or are degraded by mechanisms yet to be determined. Some of the Q109 mitochondria are **almost** completely devoid of detectable cristae, suggesting that these mitochondria are at a very different stage of degeneration and functional impairment compared to those in the other cell lines.

4. Fig 2 shows the structure of the granules in a Q77 neuron. What was the structure of the granules in the Q18 neurons? Were they similar but smaller?

Indeed, the structures of mitochondrial granules in both Q77 and Q18 iPSC-neurons look similarly heterogeneous but are smaller in size for the latter. We have revised Fig. 2 and the corresponding legend to show them side by side.

5. At various stages throughout the manuscript, the authors speculate about the nature of the sheet-like aggregates. They discuss the possibility that they are composed of mHTT, or that they may be degenerating mitochondria associated with lysosomes. Would it be possible to perform immunoEM with antibodies to HTT to determine whether they contain HTT?

We indeed performed extensive conventional transmission electron microscopy (TEM) experiments of chemically fixed and *en bloc* heavy-metal-stained cells using standard protocols, with on-section staining using Reynold's lead citrate and uranyl acetate. Unfortunately, we could not clearly distinguish any enlarged mitochondrial granules or sheet aggregates from stain deposits in our images. Because of this and the risk of chemically induced artifacts, we primarily use cryo-preserved samples for morphological observations as reported in this manuscript.

Line 287 – were the proteomic analyses performed on technical replicates for each of the two lines? At the end of the legend to Figure 7, it is stated that $n = 3$. What does this refer to?

Yes, $n=3$ corresponds to technical replicates. We have removed it here to avoid confusion. All the numbers of technical replicates for each of the data sets shown in our figures are reported in **Supplementary Table 1** and edited the Methods section to explicitly state this.

Minor edits required:

6. Line 63: the referencing for the presence of inclusions in mouse models and HD post-mortem brain is mixed up.

We appreciate the reviewer noting this. We have fixed the references and revised the text as below:

“ The propensity of mHTT to aggregate in neuronal cells is a hallmark of HD and leads to the appearance of large (micrometer scale) nuclear and neuritic inclusions, as seen in mouse models¹⁹ and human post-mortem brain²⁰”.

7. Line 333 – Should be Supplementary Fig 4c-f.

We thank the reviewer for catching this typo. We have made the corresponding change in the manuscript.

8. Line 363 – Fig 7d in the wrong place in the sentence

Fig. 7d has been moved to the supplement to provide space to incorporate the new GRSF1 data.

The sentence previously read:

“To computationally determine whether targeting PIAS1 would predict changes in the HD mitochondrial

proteome, we first compared the changes in the mitochondrial proteome of HD iPSC-derived neurons (Q109 vs Q18) described above with findings from our prior study of gene expression in the same type of differentiated neurons following PIAS1 knockdown³³ (**Fig. 7d**)."

We have now changed the sentence structure:

"To computationally determine whether targeting PIAS1 would predict changes in the HD mitochondrial proteome, we first compared the changes in the mitochondrial proteome of HD iPSC-derived neurons (Q109 vs Q18) described above with findings from our prior study³⁶ of gene expression in the same type of differentiated neurons following PIAS1 knockdown (**Supplementary Fig. 5a-b**)."

9. Lines 431-434 – Fig. 9b and 9d are not mentioned in the text.

We thank the Reviewer for catching this. Our revised text reads as following:

"Using quantification of the granule volumes after segmentation of the features (**Fig. 9**),..."

To make a general reference to Fig. 9, which includes subpanels Fig. 9b and 9d.

In addition, to more directly address the Reviewer's feedback, we have changed the paragraph to include the reference suggested by the reviewer.

"While granule volumes were increased for all Qn iPSC-derived neurons except Q109, granule numbers per nm³ of mitochondrial volume were not different in any of them from those in control Q18 cells (**Fig. 9b**). On the other hand, granule numbers per nm³ of mitochondrial volume were decreased in all mouse cell lines with respect to WT, including those treated with siRNAs (**Fig. 9d**)."

10. Supplementary Fig. 1: abbreviations should be defined

We have now spelled out "control" instead of "CTL" and defined all other abbreviations.

11. Supplementary Fig 2: what does the blue box represent?

The blue box was a second example simply from an area different from the red box, to highlight target areas in neurites, where a mitochondrion (denoted by the "mt" label in white letters) contains dense granules, visible at low magnification. To simplify, we made both boxes red and numbered them "1" and "2", so that it is clearer that they are two examples of the same feature.

12. Supplementary Fig 3: f is missing from the legend

Thank you for catching this typo. We have added "f" as "...cristae, and cristae junctions. f Semi-automated, neural-net based annotation..."

Reviewer #2 (Remarks to the Author):

The presented manuscript is technically sound. The statistical analyses of granules and sheet aggregates in various cell types are properly conducted. The main findings are that both granules and sheet aggregates are altered in the HD cells, which may be partially rescued by Pias1 KD or KO. Proteomic analysis points to a number of processes/pathways, yet the mechanism is still unclear. The identity of the membrane-bound compartments containing sheet aggregates (lysosomes, autophagosomes etc) could have been identified or defined by using a CLEM approach. Similarly, the hypothesis of mitochondria-lysosome interaction for the inclusion of sheet aggregates in the membrane-bound compartment could be tested using the CLEM approach.

We thank the Reviewer for the thoughtful suggestion. We have attempted some early experiments using cytoD and mitoD fluorescent tags, however we did not obtain sufficient information using CLEM. We will need more research effort to optimize these approaches with cryoET and these experiments are outside the scope of the current study.

Specific comments:

1. There are sheet-like aggregates (Poly-Gly-Ala) and other aggregates observed in neurons in previous studies, which are relevant to this study and should be discussed and cited.

Q. Guo et al. In situ structure of neuronal C9orf72 poly-GA aggregates reveals proteasome recruitment Cell, 172 (2018), pp. 696-705.e12)

Riemenschneider, H. et al. EMBO reports. e53890 2022

Gel-live inclusion of C-terminal fragments of TDP-43 sequester stalled proteasome in neurons

We thank the reviewer for suggesting these two publications and were aware of them. Since we are substantially over the recommended references limit, we had opted not to discuss these since they pertain to different aggregating biochemical species. However, we do agree that they are relevant to the current study in the context of protein aggregation and neurodegenerative disease. We have now added references to these publications.

2. Due to sample thickness, the authors can only survey structural features in the thin neurites. Would the observed changes also occur throughout the cell body?

Based on our recently acquired tomographic data, similarly enlarged granules in mitochondria and sheet aggregates in double-membrane-bound organelles are also observed in cryoFIB milled cell body of iPSC-neurons. This is part of our ongoing research that will be reported in a future publication.

3. Line153-155: Could the authors quantify the “abnormal” cristae?

This is an excellent question, however we were not able to quantify the abnormal cristae. The following is our experience towards this goal to annotate and quantify the cristae. We tried to train our neural networks to automatically segment cristae but found that they performed poorly with this type of feature. Furthermore, the missing wedge distorts cristae, making them appear as incomplete, open contours. We thus considered taking manual measurements of arbitrary sections of cristae, given that the morphological disruptions to cristae are readily apparent; however, we found this to be relatively imprecise and highly subjective, yielding results difficult to validate, adding little of value to our current conclusion. Nevertheless, we are indeed conducting active investigations where we will determine the intricacies, challenges, and limitations of the semi-automated neural-network based annotation method to use here for quantitative purposes, which are not yet generalizable to all subcellular features.

4. Fig5, to quantify the abundance of the sheet-like aggregates in autophagic organelles and compare them between different cell types, could the authors provide the percentage of autophagic organelles containing sheet-like aggregates, and also calculate the volume ratio between aggregates and autophagic organelle from different cell types?

This is a very salient question, but we do not have the type of data that would answer this question at this time. This requires further technological development in large scale data collection at various magnifications to make such quantification statistically meaningful. We are currently developing approaches to enable this type of analysis.

5. Fig S2, what is the blue box?

We have modified Fig S2 and explained in the caption.

Reviewer #3 (Remarks to the Author):

This manuscript discovered HD-related mitochondrial phenotypes in neurons using a combination of CryoET and proteomics approach. Mitochondrial and autophagosome morphology from multiple hiPSC-neuron lines and primary mouse neurons expressing human mHTT were examined in this study. Enlarged granules inside mitochondria are striking phenotype which was rescued or partially alleviated by PIAS1 KD. AI-based imaging processing also enabled the quantification of mitochondrial phenotypes which can be a useful tool for the field. Detailed comments are listed below:

1. In Page 7, the author mentioned that the mitochondrial phenotypes became more pronounced with higher polyQ length. Please provide the CAG repeat lengths for these five difference lpsc-neuron lines and provide a correlation analysis to prove the claim that higher polyQ length neurons have severer phenotypes.

The cell lines are labeled throughout the manuscript and in the supplemental data as Q_n, with “n” indicating the lengths of the CAG repeats for each cell line. Most importantly, Fig. 9 shows the quantification of granule numbers and volumes per cell line, demonstrating the aberrant phenotypes in

HD and statistical significance as described in the Methods. We have removed the sentence from the manuscript of increased phenotypes with increasing CAG/polyQ as it is hard for us to quantify the cristae phenotype, however qualitatively it is worse in the Q109, and the correlation of polyQ length to increased granule size isn't truly linear, although has a Pearson's correlation coefficient of $R = 0.3002$, and 95% confidence interval of -0.6760 to 0.8940 (see below). We now specify in the manuscript that the quantitation is provided in the AI section of the manuscript.

Also more information needs to be provided for these five iPSC lines. Are they from different patients? And basic patient information like age, sex, disease status et al.

This is a very helpful suggestion and we have added additional iPSC line information that is available in **Supplementary Table 4**.

2. Both hiPSC-neurons and mouse primary neurons showed enlarged granules and abnormal cristae in neurites. Are this abnormal morphology only present in neurites, what about cell body? Or is HD relevant changes mainly appearing in neurites supported by the literature?

This is an excellent question. The current work is focused on the neurites because cell bodies of neurons are too thick to view with our 300 keV electron microscope. To examine regions close to the cell body, which are very thick, would require cryoFIB-SEM to form a thin lamella prior to cryoET. During this investigation, we did not have access to cryoFIB-SEM. To our knowledge, there is no data to date that distinguishes mitochondrial dysfunction in the cell body versus neurites, including bioenergetic dysfunction of mitochondria in HD in neuronal cells.

3. Can this high resolution cryoET differentiate dendrites vs. axons?

Not at this time. The myelin sheath is the main identifying feature of axons, as it is absent in dendrites. However, the axons of both iPSC-derived and mouse primary neurons lack myelin sheaths. In our iPSC-derived neuron RNAseq dataset, we do not observe genes related to myelination and oligodendrocytes in abundance, FPKMs <1 for OLIG1, OLIG2 and MBP, suggesting these neurons are likely not myelinated, and others used coculture with rat Schwann cells to myelinate iPSC-derived neurons⁴⁴. Neurofilaments (NF-H/NF-M) are another marker that allows discrimination between axons and dendrites. However, finding the neurofilaments might be challenging as they may primarily be present in areas not related to the HD phenotypes that are the focus of our current study. Indeed, one of the phenotypes we report here in HD cell lines is the general scarcity of cytoskeleton components in the vicinity of HD mitochondria and double-membrane organelles with sheet aggregates. Thus, identifying neurofilaments unambiguously (e.g., differentiating them from actin) would require off-target data surveying and collection as well as subtomogram averaging of thousands of filament segments from hundreds of tomograms in areas that contain the filaments. The best way to distinguish between axons and dendrites would be to perform cryo-CLEM experiments with myelinated neurons expressing fluorescent NF-H in their axons; however, our goal in this study was to investigate HD phenotypes without transfecting the cells with artificial constructs.

4. Figure 6a shows isolated mitochondria using anti-TOM22 microbeads. Are these two images only from the Q109 HD neurons? Can the author also show the one from control neurons? The morphology differences in patient vs. ctrl neurons may influence their efficiency of being captured by the microbeads and therefore influencing their proteome composition/abundances.

We have modified Fig. 6 to include one cryoET panel displaying mitochondria from the Q109 cell line, and one panel showing mitochondria isolated from the control Q18 cell line.

5. It would be more convincing if the author can conduct proteomics on the rescued neurons compared to the HD neurons. Many of these changed proteins are high abundant mitochondrial proteins which should be detectable in even whole neuron proteomics without the need to isolate mitochondria.

The issue with testing the whole neuron proteomics could be confounded by the fact that protein import into the mitochondria is impaired and thus overall levels of mitochondrial proteins may not reflect those within the mitochondria (*i.e.*, proteins that should be imported into mitochondria may accumulate aberrantly in the cytoplasm or other compartments in HD cells). Therefore, the signal from whole-neuron proteomics alone could potentially be difficult to interpret without extensive validation experiments. Due to the scope of work involved, we intend to pursue such a line of study in greater depth in a separate, proteomics-centered report.

To address the reviewer suggestion, we plan to generate additional PIAS1 hetKO lines in the Q109 and other HD lines and carry out proteomics on isolated mitochondria to provide further insight as to why PIAS1 hetKO produces rescue in HD neurons. It will be important to use the Q109 with PIAS1 hetKO given that the original proteomics was carried out in that line. Generation of the lines and analyses will be part of future studies and a subsequent manuscript.

6. In Page 24. The author mentioned previous studies that identified mitochondrial dysfunction in HD neurons. Do authors see these relevant protein changes in the current proteomics dataset? For example, changes in OXPHOS complexes in the isolated mitochondria?

In **Supplementary Figure 4** we show that there is an enrichment of proteins that are involved in the TCA cycle and electron transport chain within the mitochondrial isolation preps. These GO annotations do not appear in the GO biological processes when looking at the differential proteins between HD and control neurons, but there are terms related to metabolism. Additionally, mitochondrial dysfunction was the top IPA pathway resulting from the HD vs control differential proteins. Proteins listed below were differential between HD vs control and contributed to that term (**Supplementary Table 3**). Oxidative phosphorylation pathway was a significant term outside of the top 10 (**Supplementary Figure 4**), but was documented in IPA pathways in **Supplementary Table 3**. We will add a summary sentence to highlight this finding in the revised manuscript.

Oxidative phosphorylation

ATP5F1C, ATP5PF, COX7A2, NDUFA8, NDUFAB1, NDUFB1, NDUFV1

Mitochondrial dysfunction

ACO2,ATP5F1C,ATP5PF,COX7A2,CYB5R3,FIS1,GPD2,GSR,HSD17B10,HTRA2,NDUFA8,NDUFAB1,NDUFB, NDUFV1,OGDH,SOD2,TXNRD2

Minor comments:

- *How many replicates (dish of neurons) were conducted for each group in the proteomics experiment? In the method section for mitochondrial isolation. Also, can the author indicate how many neurons in what scale of culture dish are sufficient to provide enough isolated mitochondria for proteomics?*

Optimization of the experiment required multiple wells of neurons. The final reported results come from three technical replicates (different sets of neurons) from the same differentiation in different wells for each replicate. We have added these details to the Methods section of the revised manuscript. A total of ~18 million neurons per replicate were needed to provide enough material for mitochondrial proteomics. We culture in 6 well plates and this requires 3 plates to generate the 18 million neurons.

- *In page 14, the author mentioned 177 differentially enriched peptides with a potential 236 identities. Do you mean 177 peptides belong to 236 protein groups were changing? Is the quantification performed in the protein level or peptide level?*

Quantification was performed at the protein level; however, for some peptides, there is ambiguity in correlating to the corresponding protein identity, therefore 177 peptides could belong to 236 potential proteins. We have clarified this in the text.

- *Since the paper mainly focused on mitochondria and autophagosome morphology, maybe consider revising the title to be more specific to these two organelles or just mitochondria.*

The title is at the character limit, so we cannot add both mitochondria and autophagosomal organelles without going over the character limit. Adding just mitochondria would leave autophagosomal organelles out. Furthermore, the autophagosomal organelles are a collection of organelles, possibly including phagophores, amphisomes, and autophagosomes.

Reviewer #4 (Remarks to the Author):

Thank you very much for asking me to review this manuscript from Wu et al. This is an interesting observational report which identifies intramitochondrial aggregates in the neurites of iPSC-neurons derived from patients and from mouse HD models. The numbers of granules/aggregates in the mito populations generally increased as a function of polyQ repeat, although this correlation was not complete. Surprisingly, these unusual aggregates could be partially ameliorated by depletion of a sumoylating protein PIAS1. Whilst these observations are important and should be published I feel the

paper is rather phenomenological and is a bit frustrating. The authors report these interesting series of aggregates/granules and make some suggestions as to what they may be (RNA granules ?, CaP precipitates ?) but do not follow this up. I agree this is not trivial, but it would strengthen the paper substantially if the authors knew what they were. I am also far from convinced by the authors overall conclusion that these aggregates may be due to aberrant mito protein import, as the authors suggest an association between mHTT and a component of the TIM complex which could lead to aberrant import. The authors seem to have data that contradicts this suggestion, as differentially expressed proteins identified by MS show very little evidence of a tendency towards loss of mito matrix proteins. The authors suggest that the increase in granules may actually be indications of increased RNA granules, although a key member of the RNA granule, GRSF1 is actually downregulated. As I mentioned, identifying what constitutes these granules is really important. Determining whether they are due to increased RNA aggregation should be very simple, as the authors could just perform a mitochondrial RNA:FISH expt and visualize by standard confocal microscopy. There are only 11 mt-mRNA species and 2 mt-rRNA species so it is quite simple to assess. Overall, the authors are no doubt and understandably very proud of their methodology, but I feel the paper is driven more by this methodology rather than the authors being driven by trying to identify what these aggregates are and how they are formed.

Our study here consistently identified previously unobserved early phenotypes in several HD model systems, spanning multiple HD patient-derived cell lines, as well as two mouse models (plus a third as control). In addition, we identified a promising therapeutic candidate to alleviate these phenotypes and presented preliminary data hinting at the potential biochemical compositions of the aberrant aggregates we observed in the organelles.

We agree that more conclusive results as to the biochemical identities of the enlarged granules in mitochondria and the sheet aggregates, we observed in double-membrane-bound organelles would be an important addition to the novel HD phenotypes and the technical capabilities we demonstrate here. We also recognize the apparent disconnect between the granules likely mitochondrial RNA granules (MRGs) and the fact that in the Q109 line, the proteomic data shows a downregulation of GRSF1. However, in this line, there is such profound dysmorphology of the mitochondria that most neurons do not even have granules and have significantly disrupted cristae, therefore it is not entirely unexpected that GRSF1 might be downregulated. We have clarified this point in the manuscript

However, in response to this question and given that it is difficult to directly relate FISH outputs to cryoET detection of the granules, we conducted an additional KD experiment to attempt to address whether the enlarged granules in the mitochondrial are indeed mitochondrial RNA granules (MRG). We reasoned that if we reduced expression of a critical MRG protein, in this case GRSF1 which regulates RNA processing by MRGs, we should disrupt the granule phenotype. If these are not MRGs, then GRSF1 KD should have no effect. We selected a line specifically and consistently had significantly enlarged granules (Q66) that was also used for the PIAS1 hetKO experiment. This is exactly what we observed that the size of the granules in the Q66 line was reduced. This data suggests, but does not conclusively provide, that the granules are likely MRGs.

Additional experiments to determine the identities of the aggregates and the organelles that harbor them will be developed further in future studies; however, to carry out additional technically challenging experiments, and documenting the results from them in detail and with statistical confidence, will require another full study of its own, or multiple studies and manuscripts, in future years.

SUM: As mentioned in the response to other Reviewers, we have also added Supplementary Figs. as well as additional text to the Results, Discussion and Methods sections of our manuscript to describe failed experiments that aimed at delivering more clarity.

References Cited

1. Davies, S. W. et al. Formation of neuronal intranuclear inclusions underlies the neurological dysfunction in mice transgenic for the HD mutation. *Cell* 90, 537–548 (1997).
2. DiFiglia, M. et al. Aggregation of huntingtin in neuronal intranuclear inclusions and dystrophic neurites in brain. *Science* 277, 1990–1993 (1997).
3. Morozko, E. L. et al. PIAS1 modulates striatal transcription, DNA damage repair, and SUMOylation with relevance to Huntington’s disease. *Proc. Natl. Acad. Sci. U. S. A.* 118, (2021).
4. Clark, A. J. Establishing Myelinating Cocultures Using Human iPSC-Derived Sensory Neurons to Investigate Axonal Degeneration and Demyelination. *Methods Mol. Biol.* 2143, 111–129 (2020).

REVIEWERS' COMMENTS

Reviewer #1 (Remarks to the Author):

The authors have satisfied the concerns of this referee

Reviewer #2 (Remarks to the Author):

The reviewer appreciate the technical challenges in this project. Some questions might not be readily addressable currently. Since authors were able to observe enlarged granules in mitochondria and sheet aggregates in double-membrane-bound organelles in cryoFIB milled cell body of iPSCneurons, it is recommended that the authors include these recently acquired tomographic data to strengthen the paper.

Reviewer #3 (Remarks to the Author):

The author has addressed most of my comments. Some minor comments are listed below:

1. There are a lot of ER proteins identified in the isolated mitochondrial fraction in Figure 6C. The author mentioned in line 346 that this may reflect mitochondrial-ER interaction. This conclusion can be overreaching. The data seems more like that mitochondrial isolation is not pure and with some ER contamination. Is there ER proteins that are known to not interact with mitochondrial and also show up in the mitochondrial fraction? The authors need to be careful with the discussion about ER-mitochondrial interactions because this assay cannot directly evaluate such relationship.
2. The limitations of this study should also be clearly stated in the discussion. The authors need to be careful without overdrawing conclusions.
3. The identification of 177 from 236 proteins seems odd. Can the author provide a supplemental table for these peptide and protein identification and quantification information? Protein identification in a typical proteomics experiments are based on unique peptides belong to the protein. Therefore, if a

peptide belongs to several proteins, it typically does not count as unique proteins and therefore not used for any protein identification. Unless this represents different protein isoforms.

4. Please make sure all data are publicly available. Many datasets right now remain private.

Reviewer #4 (Remarks to the Author):

The authors have made an impressive attempt at answering many of the multitude of questions that were raised by the referees, including myself. They have performed a nice expt to show that depletion of GRSF1 leads to a decrease in the size of mitochondrial granules, supporting their claim that the granules may be MRGs. I am not an expert in HD and it is clear that there are many aspects of this work that are really strong. My expertise is mitochondrial biology, however, and I still really struggle with extrapolations that the authors make from their data regarding putative effects on the mitochondria and networks. If there really is a problem with mitochondrial protein import then it should be possible just to show depletion of many mito proteins by comparing directly the proteome between wild type and HD cell lines, or even simply by comparing western blots standardised against cytosolic markers. Comparing mitochondrial proteomes seems to me to be making things more complicated. I also spent a lot of time trying to understand the section on the mitochondrial proteome in the results section and the relevant figures. I think there must have been a confusion with Fig7c and the fig legend - are these proteins in red/blue enhanced or depleted? The legend says 'Ingenuity pathway analysis of the differentially enriched proteins between HD and control mitochondria highlighted this as the top network'. What is 'this'? Yes, there are cytosolic RNA binding proteins that are modulated, but this is the mitochondrial proteome that is being compared. Please explain that these are contaminants, for example. I guess the MRPS/L that are shown are RNA-binding components of the mitochondrial ribosome. Perhaps this could be made more clear? The authors to try and reiterate when discussing the various cell lines and the granules that in some cases the granules are larger but there are far fewer of them.

Response to Reviewer comments NCOMMS-22-11340A

We greatly thank the reviewers for their positive comments.

Reviewer #1 (Remarks to the Author)

The authors have satisfied the concerns of this referee

Reviewer #2 (Remarks to the Author)

The reviewer appreciates the technical challenges in this project. Some questions might not be readily addressable currently. Since authors were able to observe enlarged granules in mitochondria and sheet aggregates in double-membrane-bound organelles in cryoFIB milled cell body of iPSCneurons, it is recommended that the authors include these recently acquired tomographic data to strengthen the paper.

The conclusions drawn from our cryoET data and its AI-based quantitative analysis of the granule features in mitochondria and sheet aggregates in double-membrane bound organelles in the neuronal processes of the human iPSC and mouse model derived neurons are rigorous, systematic, statistically adequate and compelling. We do not think that cryoFIB milled cell bodies would add necessary evidence to substantiate our claim of discovering new structural signatures in the neuronal processes of HD model neurons. In addition, we are still in the process of conducting more thorough data collection and analysis of those samples. The current data set does not yet have sufficient statistics for different cell line neurons that have the same rigor and quality for cryoFIB as those we presented in our current manuscript for the neurites after a period of three years of work.

Reviewer #3 (Remarks to the Author)

The author has addressed most of my comments. Some minor comments are listed below:

1. There are a lot of ER proteins identified in the isolated mitochondrial fraction in Figure 6C. The author mentioned in line 346 that this may reflect mitochondrial-ER interaction. This conclusion can be overreaching. The data seems more like that mitochondrial isolation is not pure and with some ER contamination. Is there ER proteins that are known to not interact with mitochondrial and also show up in the mitochondrial fraction? The authors need to be careful with the discussion about ER-mitochondrial interactions because this assay cannot directly evaluate such relationship.

This is a great point and we have removed the discussion relating to ER-mitochondrial interaction. From Cryo-ET images, there does appear to be an ER-mito interaction but we will carry out future investigations to delve into this more fully.

2. The limitations of this study should also be clearly stated in the discussion. The authors need to be careful without overdrawing conclusions.

We have provided additional discussion of limitations of the study in the revised text.

3. The identification of 177 from 236 proteins seems odd. Can the author provide a supplemental table for these peptide and protein identification and quantification information? Protein identification in a typical proteomics experiments are based on unique peptides belong to the protein. Therefore, if a peptide belongs to several proteins, it typically does not count as unique proteins and therefore not used for any protein identification. Unless this represents different protein isoforms.

We appreciate the reviewer concern and have modified the description of the proteomics analysis and carried out new analysis to only reflect unique proteins. There are now 124 unique proteins analyzed and Figures 6 and 7 modified accordingly. We have also corrected Supplementary Figures 4 and 5.

4. Please make sure all data are publicly available. Many datasets right now remain private.

We have deposited to EMDDB with accession numbers for our cryoET 3D reconstructions of both mitochondria and autophagic organelles of each iPSC neuron and mouse neurons shown as 2D slices in the manuscript. The proteomics dataset is under embargo until publication. However, the reviewer can access the dataset using the following credentials

Username: reviewer_pxd037526@ebi.ac.uk

Password: s55cQfTb

Reviewer #4 (Remarks to the Author)

The authors have made an impressive attempt at answering many of the multitude of questions that were raised by the referees, including myself. They have performed a nice expt to show that depletion of GRSF1 leads to a decrease in the size of mitochondrial granules, supporting their claim that the granules may be MRGs. I am not an expert in HD and it is clear that there are many aspects of this work that are really strong. My expertise is mitochondrial biology, however, and I still really struggle with extrapolations that the authors make from their data regarding putative effects on the mitochondria and networks. If there really is a problem with mitochondrial protein import then it should be possible just to show depletion of many mito proteins by comparing directly the proteome between wild type and HD cell lines, or even simply by comparing western blots standardised against cytosolic markers. Comparing mitochondrial proteomes seems to me to be making things more complicated.

We thank the reviewer for the comment and went back to evaluate the mitochondrial proteome further. No mitochondrial encoded proteins were depleted – only nuclear encoded proteins that require import into the mitochondria were depleted. We did the purification because we were trying to understand the nature of the granules, not knowing initially what they were. We wanted to separate the mitochondria away from the other cellular components. Also, without purification of mitochondria it is possible that one would see steady state levels of nuclear encoded mitochondrial proteins and would not have been able to detect the differences in proteins present in the mitochondria.

I also spent a lot of time trying to understand the section on the mitochondrial proteome in the results section and the relevant figures. I think there must have been a confusion with Fig7c and the fig legend - are these proteins in red/blue enhanced or depleted ? The legend says 'Ingenuity pathway analysis of

the differentially enriched proteins between HD and control mitochondria highlighted this as the top network'. What is 'this' ?

We thank the reviewer for noting the discrepancy in Fig 7c legend. We did have that in reverse and have now corrected the legend. We have also clarified what “this” referred to – now refers to the top networks as posttranslational modification, RNA, posttranscriptional modification and protein folding.

Yes, there are cytosolic RNA binding proteins that are modulated, but this is the mitochondrial proteome that is being compared. Please explain that these are contaminants, for example. I guess the MRPS/L that are shown are RNA-binding components of the mitochondrial ribosome. Perhaps this could be made more clear ?

There are RNA binding proteins that are imported into the mitochondria – for instance GRSF1 is an RBP that is imported. There are likely RBPs that are found both in the mitochondria and cytosol. We have tried to clarify the results. In contrast, the mitochondrial encoded proteins are not altered in the proteomic data of purified mitochondria.

The authors to try and reiterate when discussing the various cell lines and the granules that in some cases the granules are larger but there are far fewer of them.

We have clarified further relating granule size and numbers to the various cell lines.